# Tailoring chemical composition of solid electrolyte interphase by selective dissolution for long-life micron-sized silicon anode

Yi-Fan Tian[1,2,6], Shuang-Jie Tan[1,6], Chunpeng Yang [3], Yu-Ming Zhao[1], Di-Xin Xu[1,2], Zhuo-Ya Lu[1,2], Ge Li[4], Jin-Yi Li[4], Xu-Sheng Zhang [1,2], Chao-Hui Zhang[1,2], Jilin Tang[2,5], Yao Zhao [2,5], Fuyi Wang [2,5], Rui Wen [1,2], Quan Xu[4] & Yu-Guo Guo [1,2] ✉

Micron-sized Si anode promises a much higher theoretical capacity than the traditional graphite anode and more attractive application prospect compared to its nanoscale counterpart. However, its severe volume expansion during lithiation requires solid electrolyte interphase (SEI) with reinforced mechanical stability. Here, we propose a solvent-induced selective dissolution strategy to in situ regulate the mechanical properties of SEI. By introducing a high-donor-number solvent, gamma-butyrolactone, into conventional electrolytes, low-modulus components of the SEI, such as Li alkyl carbonates, can be selectively dissolved upon cycling, leaving a robust SEI mainly consisting of lithium fluoride and polycarbonates. With this strategy, raw micron-sized Si anode retains 87.5% capacity after 100 cycles at 0.5 C (1500 mA g$^{-1}$, 25°C), which can be improved to >300 cycles with carbon-coated micron-sized Si anode. Furthermore, the Si‖LiNi$_{0.8}$Co$_{0.1}$Mn$_{0.1}$O$_2$ battery using the raw micron-sized Si anode with the selectively dissolved SEI retains 83.7% capacity after 150 cycles at 0.5 C (90 mA g$^{-1}$). The selective dissolution effect for tailoring the SEI, as well as the corresponding cycling life of the Si anodes, is positively related to the donor number of the solvents, which highlights designing high-donor-number electrolytes as a guideline to tailor the SEI for stabilizing volume-changing alloying-type anodes in high-energy rechargeable batteries.

The ever-increasing demand for high-energy-density batteries calls for new anode materials beyond graphite used in lithium-ion (Li-ion) batteries[1–3]. Alternative electrode materials with high theoretical specific capacity are being intensively discussed, such as Li metal anode and alloying-type anode materials like silicon (Si)[4,5]. Despite the different electrochemical mechanisms of these high-capacity electrodes, their morphology and structure change hugely during charging/discharge cycling, which necessitates solid electrolyte interphase (SEI)

with various properties, particularly mechanical robustness to alleviate the volume change. For the Si anode, its advantage of ultra-high theoretical specific capacity (3579 mAh g$^{-1}$ for Li$_{15}$Si$_4$) is concomitant with the massive volume change and interfacial issues. In particular, industrial-favored micron-sized Si particles, which exhibit cost advantage and high Coulombic efficiency (CE) compared with the nanoscale counterparts[6–11], suffer from even more stringent volume change issue during cycling. Hence, it is imperative to enhance the

---

mechanical stability of the SEI against disintegration for long-life micron-size Si anodes.

Recently, many efforts have been made to mechanically stabilize the SEI on Si anodes with mainly two prevalent strategies. One is the surface coating on Si anodes[12–17]. Ex situ-fabricated coating layers (also considered as artificial SEI layers) usually feature strong mechanical properties in the initial stage, but are irreformable and inevitably suffer from dissolution and cracking during the electrochemical cycling process, which will finally lead to exposure of the active electrode surface to the electrolyte. Another strategy is to modulate the native SEI by the comprehensive design of the electrolytes, including salt-concentrated electrolytes[18], fluorinated electrolytes[19,20] or the introduction of additives[21,22]. In comparison with the irreformable ex situ-fabricated surface coating layers, the native SEI forms better contact with the electrode and may reform simultaneously upon the Si cracking[23]. Nevertheless, the native SEI are complex, usually containing inorganic components (e.g., LiF, LiOH, and $Li_2CO_3$) and organic components (e.g., various lithium alkyl carbonates and polycarbonates)[24,25]. Depending on the resilience of these components, which is correlated to the elastic modulus and strain limit, some components with high modulus (such as LiF) are desired to the mechanical strength and interface stability of the Si anode, whereas some with low resilience (either low modulus or low

strain limit) are undesired and adverse to the mechanical properties of the SEI (see Supplementary Note 1 and Supplementary Table 1)[26]. Therefore, it is urgent to design and develop a new strategy to construct SEI with desired components with favored mechanical properties.

In this study, we propose a solvent-induced selective dissolution strategy to optimize the components of SEI of micron-sized Si anodes for stable cycling performance (Fig. 1a). We use a high-donor-number (DN) solvent, gamma-butyrolactone (GBL), to selectively dissolve most undesired low-resilience compositions of SEI (e.g., lithium ethylene dicarbonate (LEDC), lithium ethyl carbonate (LEC), lithium fluorophosphates ($Li_xPF_yO_z$) and other oligomers) while keeping the tough LiF and elastic polycarbonates, which feature low solubility in GBL, for resilient SEI. The resulting selectively dissolved SEI (denoted as SD-SEI) is mainly composed of robust inorganic-polymeric components, hence capable of sustaining the volume variation of micron-sized Si anodes for stable cycling performance. We also studied the classic SEI (denoted as c-SEI) formed on micron-sized Si anodes using ethylene carbonate-based (EC-based) electrolytes and LiF-rich SEI (denoted as F-SEI) derived from propylene carbonate-based (PC-based) electrolytes as control experiments, which are found to be brittle and fragile. The micron-sized Si anode with SD-SEI in the GBL-based electrolyte displays an exceptional cycling performance (delivering a specific capacity of 1804.1 mAh g$^{-1}$

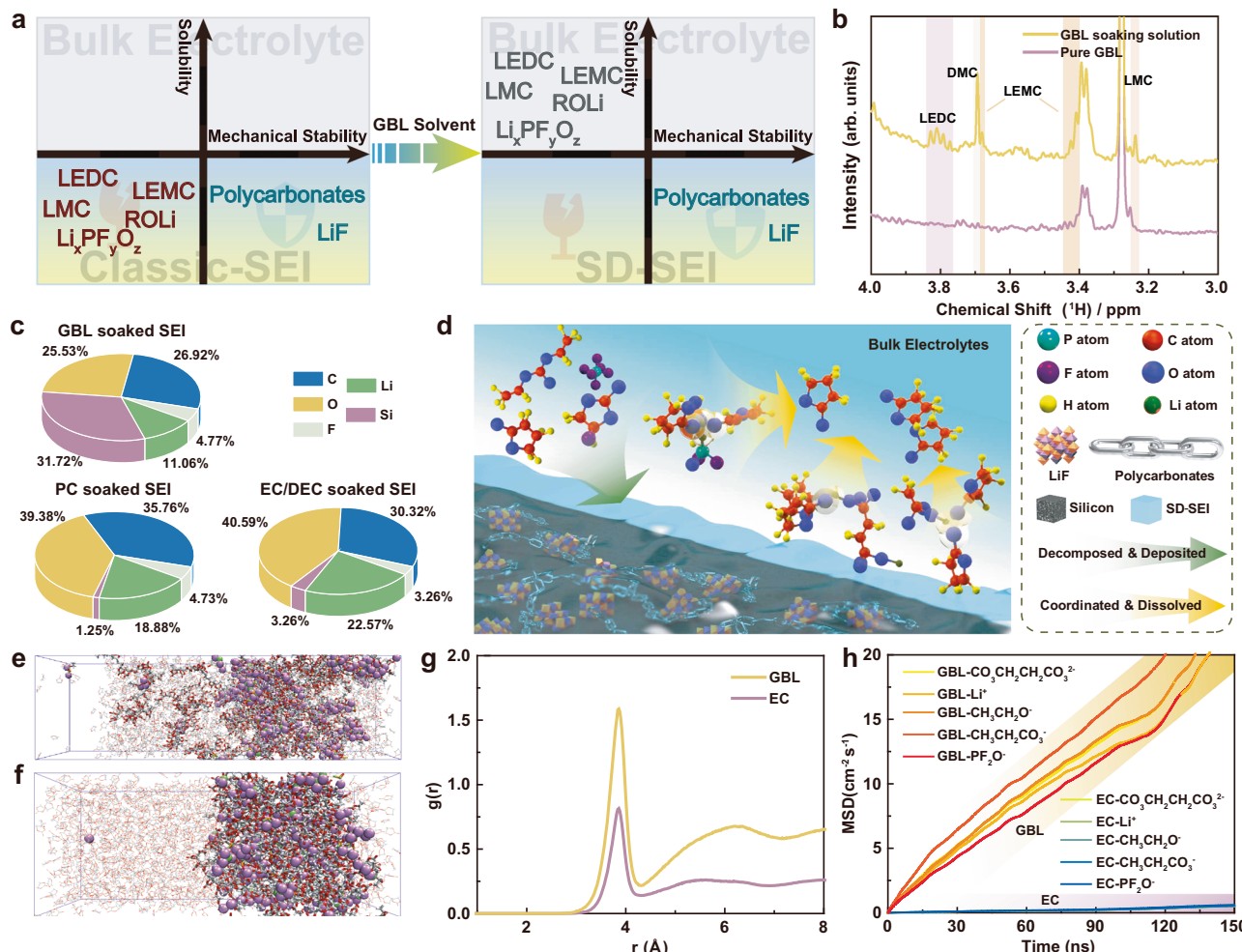

**Fig. 1 | Solvent-induced dissolution effect for SEI components. a** Schematic of solvent-induced selective dissolution for SEI, where the SEI components are screened based on their solubility in solvents. **b** NMR spectra of pure GBL solvent and GBL after soaking the cycled Si electrodes. **c** Relative elemental content of Si electrodes (analyzed by XPS spectra) that were electrochemically cycled and then soaked with various solvents. **d** Schematic of the formation process of SD-SEI in the GBL-based electrolyte. **e, f** The snapshots of MD simulation boxed showing typical

SEI species (LEDC, lithium ethoxide, LEC, and lithium fluorophosphates) in GBL solvents (**e**) and EC solvents (**f**), where the molecules of SEI species are depicted by ball-stick models and the GBL and EC molecules are illustrated by wireframes. **g** Li-O g($r$) in GBL and EC solvents. **h** The MSD analysis of $CO_3CH_2CH_2CO_3^{2-}$ (anion of LEDC), Li$^+$, $CH_3CH_2O^-$, $CH_3CH_2CO_3^-$ (anion of LEC), and $PF_2O^-$ ions in GBL and EC solvents. Source data in **b**, **g**, **h** are provided as a Source data file.

after 200 cycles at 0.2 C, where 1 C = 3000 mA g$^{-1}$), indicating excellent tolerance of the SD-SEI against the volume change. Moreover, Si‖ LiNi$_{0.8}$Co$_{0.1}$Mn$_{0.1}$O$_2$ (NCM811) full cells with the optimized SEI exhibit a capacity retention of 83.7% after 150 cycles at 0.2 C (based on the specific capacity at 4th cycle, 1 C = 180 mA g$^{-1}$), revealing favorable practical prospects of our strategy in tailoring the chemical components of the SEI on micron-sized Si anodes.

## Results

### Selective dissolution of SEI components

The formation of SD-SEI has two parallel physicochemical processes: the continuous generation of SEI due to the electrolyte-reduction products and the selective dissolution of partial SEI compositions by GBL. The processes cease when the SD-SEI is mainly composed of insoluble LiF and polycarbonates, which are stable to shield the active anode from the electrolyte. The selective dissolution capability of GBL were demonstrated by (1) directly confirming the dissolved unfavored SEI components in GBL solvent and (2) characterizing the undissolved SEI components on a cycled Si after soaked by the GBL solvent (see "Methods" for the details about the soak procedure). We first used $^1$H nuclear magnetic resonance (NMR) to analyze the species of classic SEI (which is formed in EC-based electrolytes on the Si surface) dissolved into the GBL-soaking solvent (Fig. 1b and Supplementary Fig. 1). The peaks at ~3.80 and ~3.23 ppm in the $^1$H NMR spectrum of the GBL soaking solvent are attributed to LEDC and LMC[24,27,28], and lithium ethylene mono-carbonate (LEMC) is also detected with two peaks at 3.68 ppm and 3.40 ppm[24]. The presence of LEDC and these carbonate derivates evidences the dissolution of LEDC by GBL. In contrast, no new peak is found in the NMR spectra of the PC and EC/DEC solvents after soaking (Supplementary Fig. 1), indicating the PC and EC/DEC are incapable of dissolving these undesired SEI components. The presence of the dissolved species like LEDC and LEC leads to solvated Li ions in the solvent, which were detected as charge carriers through electro-chemical impedance spectroscopy (EIS) in Supplementary Fig. 2. The ion conductivity of GBL soaking solution is $3.81 \times 10^{-8}$ S cm$^{-1}$, two magnitude higher than those of PC and EC/DMC. These experiment results prove the capability of GBL to dissolve some components of the SEI (LEDC, LMC, LEMC, etc. according to NMR), and the undissolved species were further investigated.

We further used X-ray photoelectron spectroscopy (XPS) to analyze the undissolved SEI composition on the electrode surface soaked by GBL, PC, or EC/DEC. According to the F 1s spectra (Supplementary Fig. 3a), all the SEI layers share the same peak at 685.0 eV regardless of the three types of solvents, which proves that LiF is still maintained on the electrodes after solvent soaking[29]. Different from the results of PC and EC/DEC, the XPS peaks of P-F and ROLi on the GBL-treated electrode are quite weak (Supplementary Fig. 3a, b), whereas the relative content of Si is very high (Fig. 1c), mainly because SEI was partially dissolved into the GBL solvent after soaking in GBL. Therefore, as illustrated in Fig. 1d, GBL can selectively dissolve the poor-mechanical-property components (e.g., LEDC, LEC and Li$_x$PF$_y$O$_z$) to form a robust SD-SEI, well meeting our idea of optimizing the SEI by tailoring its components. The selective dissolution effect is further investigated based on its impact on the electrochemical performance of SEI. When excluding the fluoroethylene carbonate (FEC) and LiPF$_6$ (the precursors of LiF and polycarbonates) from the GBL-based electrolyte, the formed SEI suffers from continuous dissolution and failure. Such an unstable interface gives rise to non-passivated anode surfaces and severe self-discharge (see Supplementary Note 2 and Supplementary Fig. 4)[26]. Once the additives and LiPF$_6$ were involved in electrolyte formulation (namely, the GBL-based electrolyte), a stable SEI featuring a mitigated self-discharge problem can be formed. This result demonstrates the feasibility of our selective dissolution strategy and its positive effect on the electrochemical performance of the SEI.

We conducted molecular dynamics (MD) simulation to understand the selective dissolution process of GBL. The mixing processes of the classic SEI components, including lithium fluorophosphates (LiPF$_2$O), LEC, lithium ethoxide and LEDC, in GBL and EC solvents, respectively, are illustrated in Supplementary Figs. 5 and 6 and summarized in Fig. 1e, f. After 150 ns, these SEI components mutually diffuse with the GBL solvent molecules, whereas segregated distribution is remained for EC molecules and various components, indicating the excellent dissolving capacity of the GBL toward these SEI components over EC.

The solvation structures of Li ion in GBL and EC are further compared by the radial distribution functions (g(r)) in Fig. 1g. The intensity of the first peak in g(r) for O-Li$^+$ pair is stronger in GBL in contrast to EC, which also suggests that the Li ions in the investigated classic SEI components are more likely to dissociate from the SEI and coordinate with the GBL molecules, forming a homogenous solution. Moreover, according to the mean square displacement (MSD) calculation (Fig. 1h and Supplementary Fig. 7), the Li ions and corresponding anions of the aforementioned SEI species present a much higher diffusion coefficient in GBL than in EC. The accelerated diffusion further indicates a favored dissolution process of these SEI components into the GBL solvent. In addition, the binding energy of Li-F bonds in GBL and EC is much larger than Li-solvents or F-solvent bonds (Supplementary Fig. 8), implying the high energy barriers of dissolving LiF in both GBL and EC. Thus, the theoretical simulations demonstrate the selective dissolution function of GBL, in good agreement with the experiment results.

### Inorganic-polymeric SD-SEI

The capability of GBL-based electrolyte in tailoring the SEI composition was further demonstrated by comparing the SEI layers on the Si anodes obtained after 20 cycles in different electrolytes, after which the CE reached a relatively stable state. Considering the sensitivity of SEI to the electron beam, SD-SEI was investigated by cryogenic transmission electron microscopy (cryo-TEM). The cryo-TEM image of cycled Si anodes (Fig. 2a) shows dark contrast, and the edge of the anodes with lighter contrast corresponds to the SD-SEI region (due to the lower atomic numbers of SEI components than Si). The thickness of SD-SEI is 38.0 nm based on the distinguishable edge of the Si to the outer boundary (outlined by white dashed lines in Fig. 2a). Notably, the SD-SEI contains small grains embedded in the amorphous matrix (Fig. 2b), and the lattice spacing varied from 0.77 nm to 1.27 nm are assigned to polycarbonates[30,31]. Corresponding electron energy loss spectroscopy (EELS) analysis of the Li K-edge and F K-edge from the SD-SEI (Fig. 2c and Supplementary Fig. 9a) reveals the presence of LiF, and the polycarbonates were also detected in C K-edge in Supplementary Fig. 9b. These results indicate the presence of polymeric and inorganic species in the SD-SEI. Moreover, compared with the XPS spectra of the F-SEI and c-SEI (Supplementary Fig. 10), the C 1s spectrum of SD-SEI features a distinct peak of polycarbonates at 290.2 eV, as well as a dominated signal from LiF and weak signal from Li$_x$PO$_y$F$_z$ and Li$_x$PF$_y$ in F 1s and Li 1 spectra. These peaks further evidence the presence of polycarbonates and LiF in SD-SEI with suppressed undesired species contents, which is attributed to the selective dissolution function of the GBL solvent. In contrast, the F-SEI contains only a large amount of LiF, and the c-SEI exhibits accumulating organic species, which could not be removed even if 10% FEC (by volume) was incorporated into the EC-based electrolyte (denoted as the EC-FEC electrolyte, see XPS analysis in Supplementary Fig. 11). These results further highlight the selectively dissolution strategy in screening the SEI components.

The polymeric species in SEI are further detected by matrix-assisted laser desorption/ionization time of flight mass spectrometry (MALDI-ToF-MS) for its mild ionizing function that maintains the integrity of the carbon chains. The peaks below 500 Da of the MALDI-

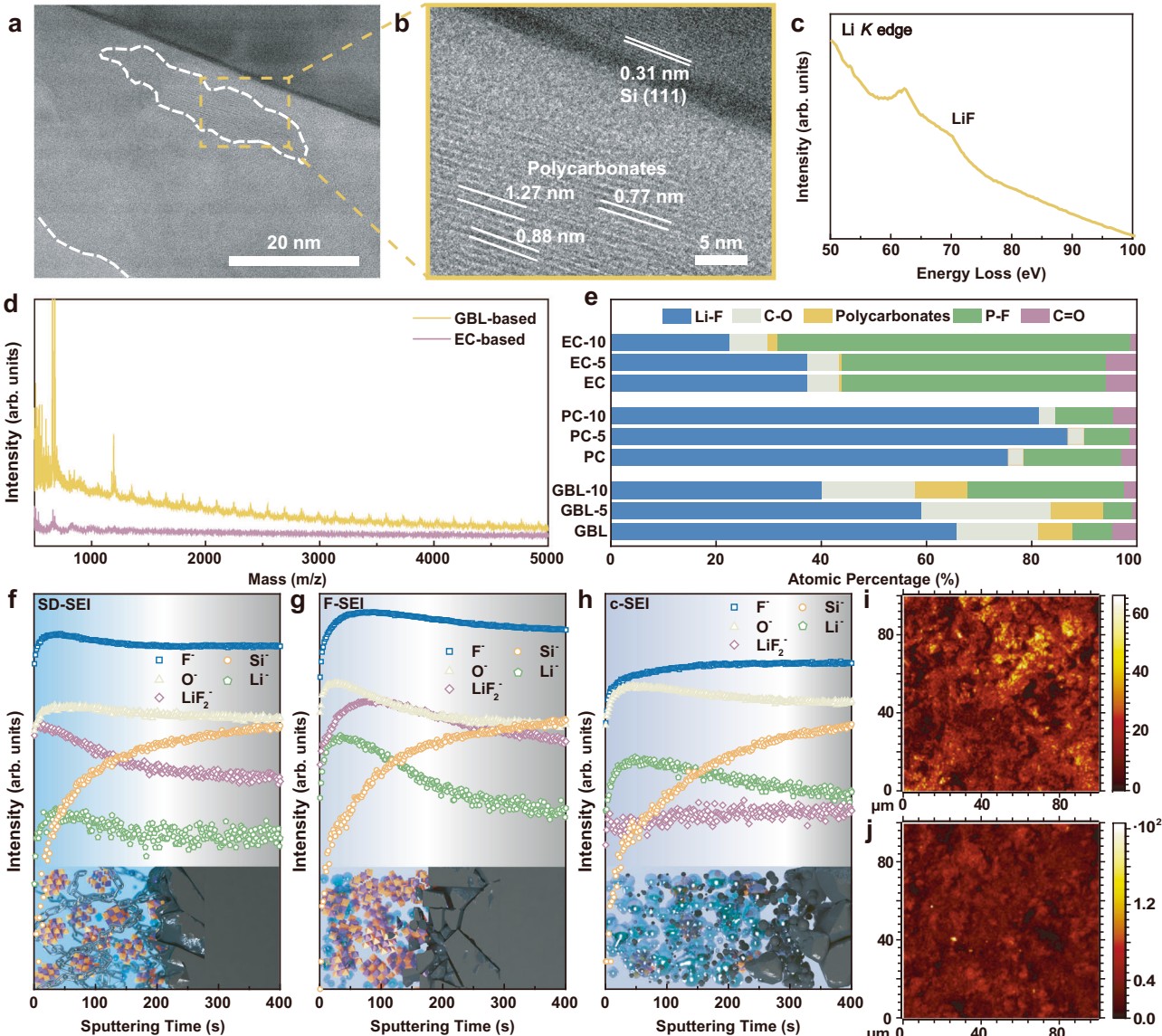

**Fig. 2 | SEI chemical composition. a**, **b** Cryo-TEM image and magnified image of SD-SEI. **c** Typical Li *K*-edge EEL spectrum of SD-SEI. **d** MALDI-ToF-MS of SD-SEI and c-SEI. **e** Relative content of selected components in the various SEI at different depths. The numbers closed to the ordinate represent the etching time (min). **f**–**j** Depth profiles of various secondary ion species obtained by sputtering in GBL-based electrolyte (**f**), PC-based electrolyte (**g**) and EC-based electrolyte (**h**), as well as ToF-SIMS secondary ion images of the cycled micron-sized Si electrodes of LiF$_2^-$ (**i**) and O$^-$ (**j**) in GBL-based electrolyte. Source data in **c**–**h** are provided as a Source data file.

ToF-MS spectra for both samples (Fig. 2d) are mainly originated from the matrix and relevant clusters. The periodic peaks appearing in the higher mass region indicate the presence of polymeric species. These polymer species are not from residual polymeric binders, as no distinct peaks show up above 1000 Da in the MALDI-ToF-MS spectrum of c-SEI. Therefore, the existence of polymers in SD-SEI is confirmed, which exhibit high molecular weights (>5000 Da, namely the maximum dictation limits of the MS).

We further analyzed the relative content of the SEI components by Ar$^+$ sputtering depth resolving technology (Fig. 2e). In contrast to the F-SEI and c-SEI containing mainly either LiF or lithium fluorophosphates, the SD-SEI features moderate amount of LiF and polycarbonates. Notably, the signal intensity of P-F increased significantly at an etching time of 10 min. This result confirms that the outer layer of SEI is too compact to be infiltrated by the electrolyte, preserving the lithium fluorophosphates in the interior of SD-SEI despite their favorable dissolubility in GBL. The SD-SEI effectively isolates the active internal part of the SEI from the electrolyte and finally gives rise to a stabilized interface.

The content and spatial distribution of inorganic species represented by LiF in SD-SEI were carefully characterized by time-of-flight secondary ion mass spectrometry (ToF-SIMS). Figure 2f–h presents the intensities for target species of cycled Si anodes with derived SEI in various electrolytes. LiF is found to be most abundant in F-SEI but deficient in c-SEI according to the intensities of LiF$_2^-$ and F$^-$. The SD-SEI possesses a moderate LiF amount, and a lower content of LiF is observed at the inner sections. This is in good agreement with the results of in-depth XPS analysis (Fig. 2e), indicating the dissolving ability of the GBL towards the undesired species. Besides, the prevalent signals of Li$^+$ and O$^-$ in F-SEI and c-SEI were significantly suppressed in SD-SEI owing to the selective dissolution effect of the GBL solvent. The corresponding secondary ion images also exhibit the homogenous distribution of LiF in SD-SEI compared with F-SEI and c-SEI (Fig. 2i and Supplementary Figs. 12a and 13a), whereas the *O*-containing species

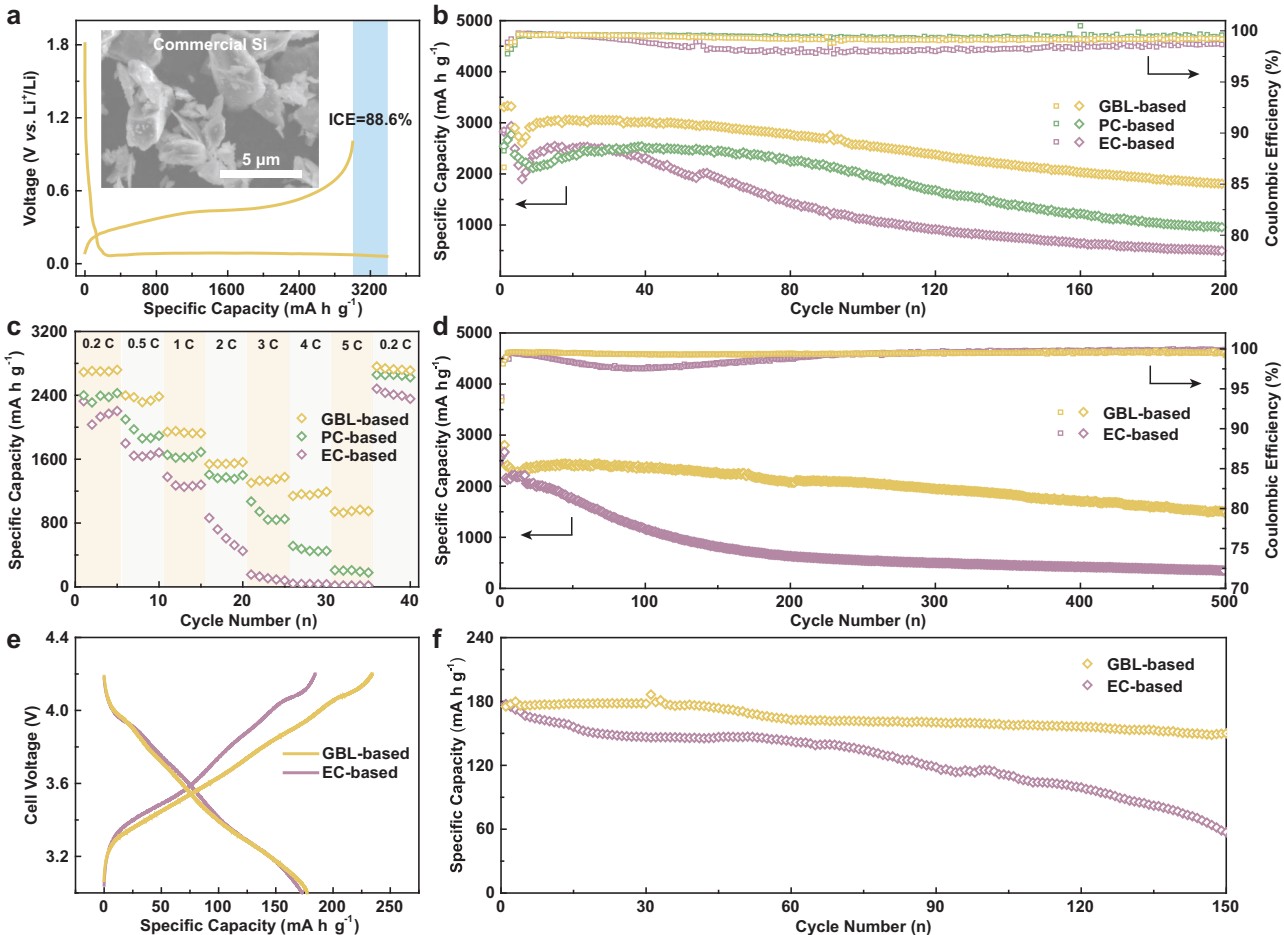

**Fig. 3 | Cycling performance of micron-sized Si electrodes in Li||Si coin cells and Si||NCM811 full-cells. a** Typical charge/discharge profile of the micron-sized Si anode in the GBL-based electrolyte at 0.05 C between 0.06 V and 1.0 V. **b** Cycling performance and CEs of the micron-sized Si anode in various electrolytes at 0.2 C. **c** Rate performance comparison. **d** Cycling performance and CEs of Si@C anode in various electrolytes at 0.2 C. 1 C = 3000 mA g⁻¹. **e**, **f** Typical charge/discharge profiles at 0.05 C between 3.0 V and 4.2 V (**e**) and cycling performance (**f**) of the Si||NCM811 full-cells using various electrolytes at 0.2 C, where 1 C = 180 mA g⁻¹. Source data in Fig. 2a–f are provided as a Source data file.

are dominant in c-SEI but rare in SD-SEI (Fig. 2j and Supplementary Figs. 12b and 13b). Thus, GBL selectively dissolves undesired species (O-containing impurities) but keeps LiF and polycarbonates to realize an ideal SD-SEI with designed components and enhanced mechanical properties.

## Electrochemical performance

The primary feature of the SD-SEI derived from the GBL-based electrolyte is its capability to stabilize anodes suffering from large volume change, which we demonstrate with a micron-sized Si anode. Commercial Si powders with an average size of 7 μm (much larger than the critical size of 150 nm as proposed by previous work[32]) were directly adopted as the anode, whose morphological and structural properties are analyzed in Supplementary Figs. 14–17. The typical charge/discharge profiles of the micron-sized Si anodes in the GBL-base electrolyte in Fig. 3a exhibit an initial capacity of 3307.2 mAh g⁻¹ in GBL-based electrolyte at 0.05 C (1 C = 3000 mA g⁻¹), higher than that of Si anodes in PC-based and EC-based electrolytes (Supplementary Fig. 18), likely due to the improved kinetics properties of SD-SEI. The initial Coulombic efficiency of micron-sized Si anode in the GBL-based electrolyte was much higher than that of nano-sized one but lower than that in the PC-based electrolyte (90.3% and 88.3% for the PC and EC-based electrolyte, respectively), which can be attributed to the charge consumption during the selective dissolution process[11]. The Si anode with the SD-SEI derived from the GBL-based electrolyte exhibits

good cycling stability, with 87.5% capacity retention after 100 cycles at 0.2 C (based on the specific capacity at 4th cycle), much higher than that of the Si anodes with PC-based (83.0%) and EC-based (44.5%) electrolytes (Fig. 3b). Such a high capacity retention can be maintained for 200 cycles, after which the Si anode in GBL-based electrolyte still delivers a specific capacity of 1804.1 mAh g⁻¹, in sharp contrast to 948.2 and 491.0 mAh g⁻¹ of the Si anodes in PC and EC-based electrolytes.

The stability is closely related to the polarization during the (de) lithiation. At the beginning of the cycling test, the similar differential capacity curves of Si anodes in various electrolytes imply an identical (de)lithiation process in Supplementary Fig. 19. Throughout the 200 cycles, the discharge voltage plateau and the plateau capacity of the anode in GBL-based electrolyte remain relatively stable compared with anodes in PC-based and EC-based electrolytes, implying enhanced cycling stability and stable SD-SEI compared to the other samples. The EIS spectra before and after 200 cycles also confirm a smaller voltage polarization of Li||Si cell with GBL electrolyte compared with PC-based electrolyte and EC-based electrolyte (Supplementary Fig. 20 and Supplementary Table 2), which confirmed the improved kinetics properties of SD-SEI. Correspondingly, the rate capabilities also demonstrate the superiority of GBL-based electrolyte and SD-SEI (Fig. 3c). In detail, the Si anodes contribute almost negligible capacity at 5 C in EC-based electrolyte, while nearly 1000 mAh g⁻¹ capacity is retained in the GBL-based electrolyte at the same rate, implying that the SD-SEI is efficient in conducting Li ion for the high rate

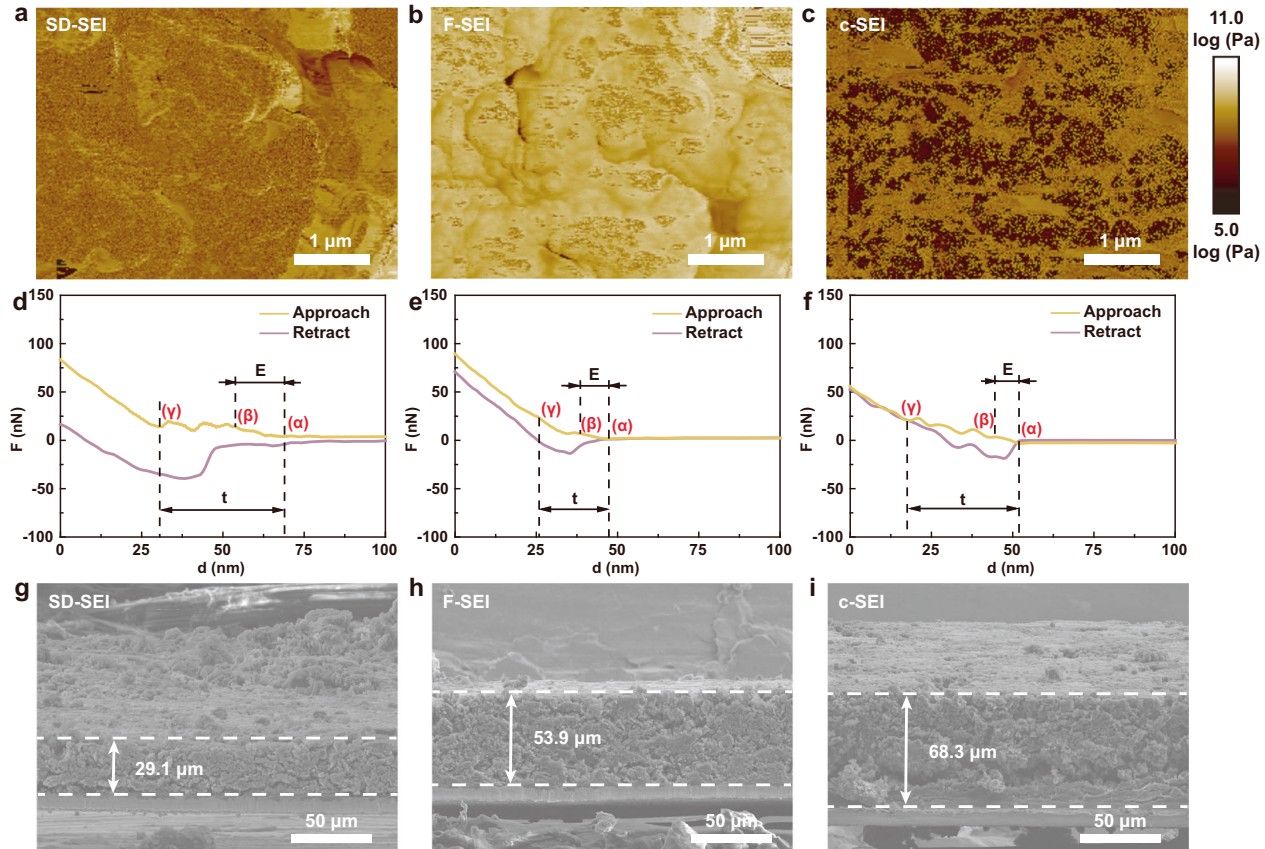

**Fig. 4 | Mechanical properties of SEI. a, b** Derjaguin–Muller–Toporov (DMT) modulus mappings of (**a**) SD-SEI, (**b**) F-SEI, and (**c**) c-SEI. **d–f** Force-displacement curves obtained on the (**d**) SD-SEI, (**e**) F-SEI, and (**f**) c-SEI. **g–i** Cross-sectional images of scanning electron microscopy (SEM) images of Si electrodes after 100 cycles in GBL-based electrolyte (**g**), PC-based electrolyte (**h**), and EC-based electrolyte (**i**). Source data in **d–f** are provided as a Source data file.

performance. The enhanced electrochemical performance of SD-SEI is owing to the selective dissolution strategy by GBL. In contrast, the incorporation of FEC without GBL cannot realized stable cycling performance in the EC-FEC electrolyte (Supplementary Fig. 21).

Furthermore, the cycling performance of micron-sized Si can be improved using carbon coating micron-sized Si (denoted as Si@C) instead of the raw Si (Fig. 3d). Nevertheless, while the carbon layer gives rise to better cycling stability to all samples, the rapid decay of Si anodes without suitable SEI layers (c-SEI) is still inevitable. In contrast, the SD-SEI facilitates the Si@C with superior cycling performance, which shows high capacity retention of 80.6% after 300 cycles (based on the specific capacity at 4th cycle). Such performance is superior than most previous works adopting micron-sized Si and even other Si-derived materials featuring enhanced structural and interfacial stability as anode[18,33–35].

Full cells made of the micron-sized Si anode coupled with NCM811 were tested to further evaluate the practical feasibility of GBL-based electrolyte and the derived SD-SEI. The compatibility of GBL-based electrolytes with NCM cathode is first verified by half cells of NCM811 (Supplementary Fig. 22). Enlightened by the compatibility, we assembled Si||NCM811 full cells using various electrolytes with N/P ratio of 1.1. Both cells displayed an initial reversible specific capacity of 175 mAh g$^{-1}$ from 3.0 V to 4.2 V at 0.05 C (1 C = 180 mA g$^{-1}$, Fig. 3e, f), but the cell with GBL-based electrolyte shows superior stability in the following cycles and maintains 83.7% capacity after 150 cycles, much higher than 44.3% of the cell with EC-based electrolyte. These above electrochemical performances, which exhibited good reproducibility according to Supplementary Fig. 23, not only validate the feasibility of GBL-based electrolyte as a practical candidate to couple with Si-based

anode materials but also support our idea of rational design of electrolyte and SEI for Si anodes. Practical conditions (especially limited electrolyte amount) are further investigated (see Supplementary Note 3 and Supplementary Figs. S24 and S25), which further demonstrates the practicability of the selective dissolution strategy and the GBL-based electrolytes.

## The robust mechanical properties of SD-SEI

The resultant electrochemical performance, especially the cycling performance of the micron-sized Si anodes reinforced by the SD-SEI is attributed to the mechanical properties of SD-SEI with the inorganic-polymeric composition. According to the atomic force microscopy (AFM), the SD-SEI is rather flat and compact in contrast to F-SEI and c-SEI (Supplementary Figs. 26 and 27). Interweaving with both highly elastic polycarbonates and stiff LiF, SD-SEI exhibits a suitable average modulus of 1.5 GPa (Fig. 4a). The F-SEI with high content of LiF shows the highest average modulus of 2.6 GPa (Fig. 4b), and the impurity-rich c-SEI has a much lower average modulus of 0.7 GPa (Fig. 4c). As confirmed by the modulus evolution collected in Supplementary Figs. 28 and 29, the modulus of SD-SEI is stable during cycling test, indicating its enhanced mechanical properties and stable structure. The nanoindentation test was also conducted to uncover the SEI evolution in response to the stress (see Fig. 4d–f, Supplementary Note 4, and Supplementary Figs. 30–32). The F-SEI is found to be thinner resulting from a large amount of high-resistance LiF, and SD-SEI and c-SEI are much thicker (20.0, 32.8, and 33.2 nm, respectively). As for the maximum elastic deformation limit[36], SD-SEI shows an overwhelming superiority to that of F-SEI and c-SEI (15, 8, and 7 nm according to the average result shown in Supplementary Fig. 33). These results

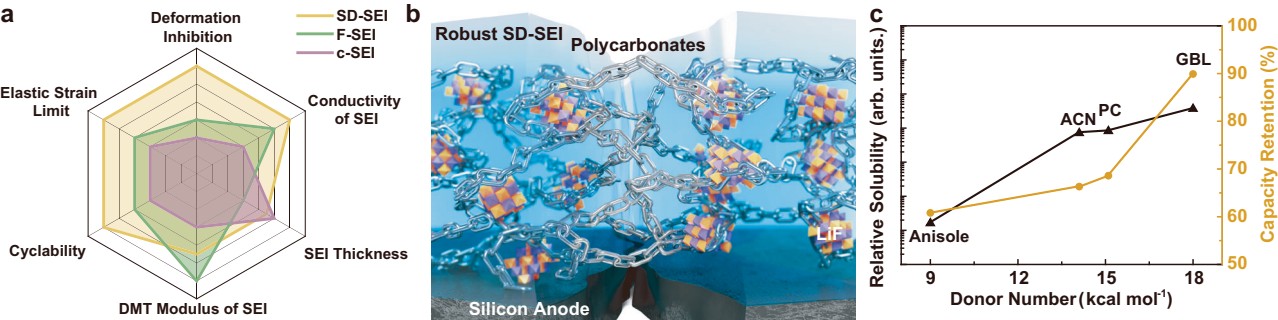

**Fig. 5 | Schematic and comparison of SEI. a** Comparison of various factors of different SEI. **b** Schematic of polycarbonates and LiF in SD-SEI, which is defending the interfacial stability against the volume change of the Si anodes and the crack of SEI. **c** The relationship of DN, relative solubility of various solvents and capacity retention of Si@C anodes in corresponding electrolytes after 100 cycles. The cycling tests was performed at 0.2 C at 25 °C (1 C = 3000 mA g$^{-1}$). Source data in **a**, **c** are provided as a Source data file.

effectively prove the comprehensive advantage of this inorganic-polymeric structure of SD-SEI from the GBL-based electrolytes in accommodating large volume expansion of the alloying type anodes.

Due to the stabilized interphase and limited side reactions, the electrode swelling is also suppressed in referring to the measurement of the anode thickness at the 100th cycle in Fig. 4g–i. Compared with the pristine state in Supplementary Fig. 34, the thickness of the Si electrode increased from 18.8 to 29.1 μm after cycling in the GBL-based electrolyte, whereas the electrodes in PC-based and EC-based electrolytes expanded to as thick as 53.9 μm and 68.3 μm, respectively. The suppressed volume change of the electrode in GBL-based electrolyte is mainly attributed to the stabilized SEI and consequently restrained interfacial side reactions, which give rise to enhanced CE and cycling performance.

Thus, the SD-SEI due to the selective dissolution of the high-DN GBL solvent affords optimized mechanical parameters, such as elastic strain limit, deformation-inhibiting capability, and other physico-chemical properties, compared with F-SEI, c-SEI, and SEIs in previous reports (Fig. 5a and Supplementary Table 3). The optimized mechanical properties benefit from the tailored components of SEI. Specifically, the SD-SEI is anchored by its tough LiF and connected by elastic polycarbonates (Fig. 5b), preserving the integrity of the SD-SEI for long-life micron-sized Si anodes[37]. The synergistic effect between LiF and polycarbonates was also reflected by the decreased cycling stability of the micron-sized Si anode with SEI of lower LiF and polycarbonates content (Supplementary Figs. 35 and 36 and Supplementary Table 4). The excellent practicality of SD-SEI is successfully extended to commercial micron-sized SiO$_x$ anodes, demonstrated by enhanced cycling performance in Supplementary Fig. 37. Furthermore, a pouch cell assembled coupling NCM811 cathode and graphite-SiO$_x$@C composite anode (500 mAh g$^{-1}$, denoted as G-SiO$_x$) exhibits stable cycling performance with 80% capacity retention after 350 cycles at current of 200 mA (based on the specific capacity at 4th cycle with first 3 cycles as formation process, see "Methods" and Supplementary Fig. 38).

The selective dissolution of SEI is closely related to the DN of the solvent of the electrolyte, as the DN functions as an index to the Lewis basicity of a solvent and indicates the tendency to coordinate with the cations and dissociate ion pairs in salts, that is, solubility[38–42]. We further used acetonitrile (ACN) and anisole as the electrolyte solvents, which show distinct DNs (14.1 kcal mol$^{-1}$ of ACN and 9.0 kcal mol$^{-1}$ of anisole, in comparison to 18.0 kcal mol$^{-1}$ of GBL), and investigated their corresponding solubility (Supplementary Fig. 39), which exerts different impact on the electrochemical performance. The Si@C anode with the ACN-based electrolyte shows enhanced capacity during the cycling compared with the anisole-based and EC-based electrolytes (Supplementary Fig. 40), which is ascribed to the robust SEI maintaining the integrality of the Si anode using the ACN-based electrolyte.

Summarizing the relationship between the DN of solvents and electrochemical performance of Si-based anodes in the corresponding electrolytes (Fig. 5c), an enhanced cyclability of Si-based anodes can be achieved when employing a high-DN solvent in the electrolyte, owing to the selective dissolution effect.

## Discussion

In conclusion, we proposed a strategy for constructing component-optimized inorganic-polymeric SEI for micron-sized Si anode by selectively dissolving unfavorable SEI components with high-DN solvents. Inspired by this strategy, some solvents represented by GBL were selected to create a high-DN environment, where massive poly-carbonates and inorganic species were deposited and maintained to build the inorganic-polymeric SD-SEI. Owing to the inorganic and polymeric main components in SD-SEI with strong mechanical toughness, the raw micron-sized Si anode realized 87.5% capacity retention after 100 cycles at 0.2 C (based on the specific capacity at 4th cycle, 1 C = 3000 mA g$^{-1}$), which can be further extended to over 300 cycles when the carbon-coated micron-sized Si was used. Si||NCM811 full-cell with GBL-based electrolyte showed high-capacity retention of 83.7% after 150 cycles at 0.2 C (based on the specific capacity at 4th cycle, 1 C = 180 mA g$^{-1}$). Furthermore, based on the comprehensive characterizations, the chemical components and structure of the SD-SEI prove the feasibility of our strategy, and the mechanical investigation supports the SD-SEI with strengthened stability in addressing the volume change of the micron-sized Si anode. This work not only offers a strategy to tailor the SEI of micron-sized Si anode, but also uncovers the relationship between the physicochemical properties (DN, solubility) of solvents and electrochemical performance. Applying high-DN solvents in electrolytes for SD-SEIs in combination with the basic requirements of electrolyte solvents, such as potential window, wettability, melting/boiling points, forms a general guidance for designing new electrolytes and tailoring SEI of alloying-type electrode materials for high-energy Li-ion batteries.

## Methods
### Characterizations

Nuclear magnetic resonance (NMR) spectra were acquired on the Bruker Avance 400 at 399.2 MHz for $^1$H, where deuterated dimethyl sulfoxide was used as the solvent. To precisely identified species dissolved in GBL, we soaked the cycled Si electrodes (after 3 cycles of 0.05 C in EC-based electrolyte, where 1 C = 3000 mA g$^{-1}$) in pure GBL, pure PC, and EC/DEC mixed solvents (mixture of 1:1 by volume was used here as the pure EC freezes at 25 °C), whereby the soluble components could be extracted and dissolved in the solvent. Typically, 20 ml corresponding solvent was applied to soak 1 piece of cycled Si electrode (~1.6 mAh). After 72 h soaking process at 25 °C, both the

soaking solutions and the soaked electrodes were characterized carefully to monitor the dissolved and retained species by the following characterizations. The chemical composition and environment were probed with an Escalab 250XI X-ray photoelectron spectroscopy (XPS) device. The nanoscale morphology and elemental mapping was investigated by field emission scanning electron microscopy (SEM, JEOL JSM-6701F) operated at 10 kV with an energy dispersive spectroscopy (EDS) system. Particle size distribution (PSD) was carried out on a Malvern Mastersizer 3000 laser particle size distribution analyzer. Raman spectrum was performed on a DXR Raman Microscope (Thermo Scientific) with a laser wavelength of 532 nm, which was first calibrated with a Si wafer ($520\ cm^{-1}$). The X-ray diffraction (XRD) patterns were collected by Rigaku D/max 2500 X-ray diffractometer with Cu Kα radiation. Matrix-assisted laser desorption/ionization time of flight mass spectrometry (MALDI-ToF-MS) measurement was performed by a Bruker UltrafleXtreme instrument. The ions composition and distribution of the anode surface and depth sputtering area were characterized by Time-of-Flight secondary ion mass spectrometry (ToF-SIMS, ToF-SIMS 5 ION-ToF GmbH, Münster, Germany). ToF-SIMS was equipped with a 30 keV $Bi_3^+$ primary ion gun and a 2 keV $Cs^+$ sputter gun, and an electron flood gun was used for charge neutralization. The sample pretreatment procedure and details in characterization were guided by previous literatures[43,44], where the SEI composition was eluted from the DMC-cleaned cycled electrode with ethyl acrylate, and the 2,5-dihydroxybenzoic acid was employed as the matrix. The cryogenic transmission electron microscopy (Cryo-TEM) characterizations were carried out using a Thermo Scientific Themis 300 operated at an accelerating voltage of 300 kV. Derjaguin–Muller–Toporov (DMT) modulus images were acquired from commercial atomic force microscopy (AFM, Bruker Multimode 8 with a Nanoscope V controller) using an insulating silicon AFM tip ($k = 26\ N\ m^{-1}$, $f_0 = 300\ kHz$) in an argon-filled glove box with peak force tapping mode, and the AFM results were analyzed by Nanoscope Analysis software. The force-displacement curves were collected by the Nano Indentor (Anton Paar, UNHT).

### Electrolyte preparation

Battery-grade reagents in the experiments were purchased and used without any further purification. To prepare GBL-based electrolyte, 1 M lithium hexafluorophosphate ($LiPF_6$, 98%, Alfa) was dissolved in the mixture of γ-Butyrolactone (GBL, 99%, Sigma-Aldrich), diethyl carbonate (DEC, 99%, Alfa) and fluoroethylene carbonate (FEC, 98%, Alfa) with volume ratio of 45:45:10. PC-based electrolyte shared the same formulation with GBL-based electrolyte, except for replacing the GBL with propylene carbonate (PC, 99%, Alfa). As for the preparation of EC-based electrolyte, 1 M lithium hexafluorophosphate ($LiPF_6$, 98%, Alfa) was dissolved in the mixture of ethylene carbonate (EC, 99%, Alfa) and DEC (1:1 by volume). The formulations of mGBL and mPC-based electrolytes were modified based on the GBL and PC-based electrolytes, where 1 M bis(trifluoromethane)sulfonimide lithium (LiTFSI, 99.0%, Sigma-Aldrich) was dissolved in the mixture of GBL/PC and DEC (1:1 by volume). Similarly, the only difference between mEC-based electrolyte and EC-based electrolyte lies in the salt type, where LiTFSI was used in mEC-based electrolyte. The EC-FEC electrolyte was prepared by dissolving 1 M $LiPF_6$ in mixture of EC/DEC/FEC (45:45:10 by volume). All the electrolytes were thoroughly stirred for 12 h before use. The whole process of electrolyte preparation was conducted in the Ar-filled glove box with water and oxygen concentration less than 0.1 ppm. The anisole-based electrolyte and ACN-based electrolytes shared formulation of GBL-based except for the GBL solvent, where anisole and ACN were used in the corresponding electrolytes.

### Molecular dynamics simulation methods

All the small molecules involved in this study were first optimized by Gaussian16 software at a level of B3LYP/def2tzvp. The optimized geometries were used to obtain the GAFF2 force field from the ACPYPE sever (https://www.bio2byte.be/acpype/)[45]. The initial structure of the systems was built with Packing Optimization for Molecular Dynamics Simulations (Packmol) program[46]. All molecular dynamics simulations were performed with the Gromacs 2019.6 program. The simulation process was detailed as following: the 5000-step steepest descent method and 5000-step conjugate gradient method were used to avoid unreasonable contact of system. NPT ensemble was used to pre-equilibrate the system, and V-rescale temperature coupling and parrinello-rahman pressure coupling were used to control the temperature to 298 K, the pressure was maintained at 1 atm, the non-bonding cutoff radius was 1.2 nm, and the integration step was 2 fs. Finally, 150 ns simulation was performed, and the bond length and angle were constrained by the linear constraint solver (LINCS) algorithm. The two-way intercept was set to 1.2 nm, van der Waals interaction, and the long-distance electrostatic interaction was set by the particle-mesh Ewald method. The trajectory file during simulation was saved every 10.0 ps. The initial and final configurations were supplied in the Supplementary Data.

### Electrochemical measurements

Electrochemical performances were evaluated in 2032-type coin cells. All the coin cells were fabricated in the Ar-filled glove box with water and oxygen concentration less than 0.1 ppm. To fabricate the micron-sized Si electrodes, a slurry was first prepared by mixing active material (micron-sized Si, 99%, Sigma-aldrich), polyacrylate composite binders (PAA), carbon black (super P) and carbon nanotube with the mass ratio of 80:10:9.8:0.2. In the following, the slurry was cast on to a copper foil, dried at room temperature for 6 h, and further dried at 60 °C for 6 h in vacuum oven. The typical mass loading of the Si anode is ~1.2 mg cm$^{-2}$ and the corresponding areal capacity is ~2.0 mAh cm$^{-2}$. Similarly, the slurry of NCM811 cathodes was prepared by mixing commercial NCM811 materials, poly(vinylidene fluoride) (PVDF) and carbon black (super P) with the mass ratio of 8:1:1. The resultant slurry was cast on the Al foil and dried at 80 °C for 12 h in vacuum oven. The typical mass loading of the NCM811 cathode is ~13 mg cm$^{-2}$ and the corresponding areal capacity is ~1.8 mAh cm$^{-2}$. In Li‖Si coin cells, the prepared micron-sized Si electrodes, Li foils (500 μm) and Celgard 2500 polypropylene membranes were used as working electrodes, counter electrodes and separators, respectively. And the electrolyte used for cell assembly included GBL-based, PC-based and EC-based electrolytes, where 50 μL electrolyte was used in each coin cell. The charge/discharge measurements for half cells measured at 0.05 C for the first 3 cycles and 0.2 C for the subsequent cycles between 0.06 V and 1.0 V (vs. $Li^+$/Li) by Neware (CT-4008T), where 1 C = 3000 mA g$^{-1}$. As for the full cell, the as-prepared NCM811 electrode was utilized as cathode, and 70 μL electrolyte was used in one Si‖NCM811 full cell. The NP ratio is about 1.1 (~1.60 mAh for anode and 1.45 mAh for cathode), and the applied voltage window was 3–4.2 V at current density of 0.2 C (1 C = 180 mA g$^{-1}$). In terms of SiO$_x$ electrodes, the SiO$_x$ (commercial materials obtained from Beijing IAmetal New Energy Technology Co., Ltd) was mixed with PAA and super P to form the slurry with the mass ratio of 8:1:1 before casting on the copper foil and dried in vacuum oven overnight. The typical mass loading of the SiO$_x$ anode is ~1.2 mg cm$^{-2}$ and the corresponding areal capacity is ~1.5 mAh cm$^{-2}$. Similar procedure was applied in analyzing the electrochemical performance of Li‖SiO$_x$ coin cells with various electrolytes except for the potential window from 0.005 V to 1.5 V (vs. $Li^+$/Li). 400 mA h G-SiO$_x$‖NCM811 pouch cell were projected and assembled with 5 pieces of G-SiO anode (mixing commercial graphite and SiO$_x$ materials, the overall specific capacity is 500 mAh g$^{-1}$) with areal capacity of 5.0 mAh cm$^{-2}$ and 4 pieces of NCM811 cathode with areal capacity of 4.6 mAh cm$^{-2}$. The electrolyte mass loading is 3 g Ah$^{-1}$. Before cycling test with current of 200 mA, 20 mA current was applied as formation process in the initial 3 cycles. The electrochemical impedance spectral

(EIS) measurements were acquired within a frequency range between 100 kHz and 0.1 Hz. During the tests above, the temperature was controlled at 25 °C.

## Data availability

The data that support the findings of this study are available in the online version of this paper and the accompanying Supplementary Information or available from the corresponding authors on reasonable request. The source data underlying Figs. 1b, g, h, 2c–h, 3a–f, 4d–f, and 5a, c, along with Supplementary Figs. 1–4, 7–11, 15–25, 27, 29, 32, 33 and 35–40 are provided as a Source data file. Source data are provided with this paper.

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

## Acknowledgements

This work was supported by the Basic Science Center Project of National Natural Science Foundation of China (Grant No. 52388201), the CAS Project for Young Scientists in Basic Research (Grant No. YSBR-058), the National Key R&D Program of China (Grant No. 2021YFB2400200), the National Science Foundation of China (Grant No. 52002374), and the Beijing Natural Science Foundation (Grant No. Z220021).

## Author contributions

Y.-G.G. conceived the projects and designed the experiments. Y.-F.T. and S.-J.T. performed the experimental work under the help of D.-X.X., J.-Y.L. and G.L. Y.-F.T. and S.-J.T. conducted the characterizations and data analysis under the help of C.Y. Z.-Y.L., X.-S.Z., C.-H.Z., R.W., and Q.X. J.T., Y.Z. and F.W. performed the TOF-SIMS characterizations. Y.-F.T., C.Y. and Y.-M.Z. co-wrote the manuscript. All the authors discussed the results and commented on the manuscript.

## Competing interests

The authors declare no competing interests.

## Additional information

[1]CAS Key Laboratory of Molecular Nanostructure and Nanotechnology CAS Research/Education Center for Excellence in Molecular Sciences, Institute of Chemistry Chinese Academy of Sciences (CAS), 100190 Beijing, P. R. China. [2]School of Chemical Sciences, University of Chinese Academy of Sciences, Beijing, P. R. China. [3]School of Chemical Engineering and Technology, Tianjin University, 300072 Tianjin, P. R. China. [4]Beijing IAmetal New Energy Technology Co., Ltd, Beijing, P. R. China. [5]CAS Key Laboratory of Analytical Chemistry for Living Biosystems, CAS Research/Education Center for Excellence in Molecular Sciences, National Centre for Mass Spectrometry in Beijing, Institute of Chemistry Chinese Academy of Sciences (CAS), Beijing, P. R. China. [6]These authors contributed equally: Yi-Fan Tian, Shuang-Jie Tan. ✉e-mail: ygguo@iccas.ac.cn

