## [Peer Review File · Nature Communications]

REVIEWER COMMENTS

Reviewer #1 (Remarks to the Author):

The manuscript describes a novel concept regarding fine-tuning of solid-electrolyte-interphases by dissolving presumably harmful components by GBL-solvent. The concept is certainly novel and highly interesting and makes a step towards purposely designed SEI-layers for improved long term performance of silicon based electrodes. Nevertheless, a publication in Nature Communications cannot be recommended, because too many questions remain unclear and the assumption that those specific components in the SEI are actually harmful is far fetched. Moreover, many uncertainties regarding the impact on the cycle life remain unclear and the main hypothesis is not well enough supported. As one example, I could not find reference spectra of the identified "undesired" components of the SEI dissolved in GBL.

Saying this, the work is of importance to the scientific community and a lot of interesting results are in the manuscript, which deserve publication in a more specialized journal focussing on electrochemistry and batteries.

Reviewer #2 (Remarks to the Author):

- What are the noteworthy results?

- The high capacity and good capacity retention are impressive for a micron-size Si-based anode. Also, the proposed method is evidently effective in modifying/regulating the SEI.

- Will the work be of significance to the field and related fields? How does it compare to the established literature? If the work is not original, please provide relevant references.

- One of the biggest hurdles in the Si community is achieving a high capacity with good capacity retention and the work presents a novel approach to address this by in-situ regulating the SEI composition.

- Does the work support the conclusions and claims, or is additional evidence needed?

- The authors have performed a comprehensive characterization of the SEI, combining computational and experimental techniques. The main claim is that the improved performance is primarily due to an SEI with good mechanical properties, achieved by dissolving the unwanted components. However, it is highly possible that the higher capacity is in part, if not mainly, due to a thinner SD-SEI compared to F-SEI or c-SEI. For instance, Fig. 1c shows more Si for the sample soaked in GBL, which is a good indication of a thinner SEI for this system. EIS, although shown only after 200 cycles, may also be an indication of a thinner SD-SEI. It would be good if the authors can explore this aspect in their discussion.

- In relation to the comments above, SD-SEI presents the highest initial capacity. Is this attributed to the mechanical property or to other characteristics of the SEI? A mechanically robust SEI does not necessarily translate to the accessible capacity of Si especially in the beginning of cycling.

- The mechanical aspect of the SEI may indeed be responsible for the cycling stability for SD-SEI. I believe that it is necessary to show characterization at various points upon cycling, e.g. cycle 1, cycle, 10, cycle 50, etc., to show that the SEI properties, particularly the mechanical integrity is indeed maintained upon cycling.

- It has been mentioned that the SD-SEI is compact. A surface image should be added to support this claim.

- Are there any flaws in the data analysis, interpretation and conclusions? Do these prohibit publication or require revision?

- The discussion emphasizes early on that LiF is a beneficial SEI component, giving the impression that the more LiF the better the SEI. However, F-SEI has the most amount of LiF but has a much inferior performance, due to the brittle and fragile nature of its SEI. On the other hand, SD-SEI has the most desirable SEI with moderate amount of LiF and polycarbonates, resulting in a tough and mechanically robust SEI. It should be highlighted that it is potentially not the absolute amount of LiF in the SEI that is important, rather its amount relative to other species such as polymeric species in the case of SD-SEI. For instance, what if the same amount of LiF is left (relative to a control) after dissolution of unwanted SEI components; would the modified SEI still perform better?

- Is the methodology sound? Does the work meet the expected standards in your field?

- The work is supported by several methods to support the claims made which therefore meet the expected standards.

- Is there enough detail provided in the methods for the work to be reproduced?

- Yes, there is enough details provided in the methods section. However, there is no mention regarding the reproducibility, particularly the electrochemical performance data.

Reviewer #3 (Remarks to the Author):

Mechanical issues surrounding SEI formed on Si are inevitable and critical for longevity of Si-based electrodes. Their concept of modifying physical properties of SEI by selectively tuning SEI's components via electrolyte additive (GBL) are highly intriguing and expected to be practical. The authors provided results from a variety of characterizations to demonstrate the advantage of their approach. However, I have several questions and concerns around their characterization approaches and deriving conclusions, especially around SEI's mechanical aspects. Since the core advantage of their approach is improved mechanical stability of SEI, I hope the authors properly address my concerns before acceptance for publication in Nature Comm. Below are my questions.

1. Authors are using three types of electrolytes. i) GBL + DEC + FEC, ii) PC + DEC + FEC, iii) EC + DEC. FEC is widely known to improve the performance of Si-based anodes. Why FEC is excluded from the EC-based electrolyte? Since PC is known to form poor SEI, to truly show the advantage of GBL over conventional electrolyte formulations, authors should compare electrochemical performance of Si electrodes between GBL + DEC + FEC vs. EC + DEC + FEC.

2. Line 117: How do the authors define 'poor-mechanical-property'? Elastic modulus? Fracture toughness? Elastic strain limit? Please provide further ground why authors consider the suggested compounds (LEDC etc.) are mechanical poor components along with either mechanical measurements or references to back it up. Also, compounds such as LEDC may be beneficial for the passivity of Si surface.

3. Lines 161-190: Daniel Abraham in ANL published several articles including 'What Makes Fluoroethylene Carbonate Different?'. In his articles, FEC promotes formation of both LiF and polymeric species. Authors discuss in this part that dissolution induced by GBL is the origin of higher LiF and polymeric species in SD-SEI compared to c-SEI. However, based on Daniel Abraham's papers, I wonder whether the observation is from the existence of FEC in GBL-based electrolyte while no existence of FEC in EC-based electrolyte.

4. Please provide references stating LEDC or other species are 'undesirable' and elaborate why they are undesirable.

5. Line 228-231: i) Please explain why GBL yields higher specific capacity. The described advantage of GBL doesn't seem to directly coupled to capacity utilization. ii) Higher coulombic efficiency is from lower surface area of micron-sized Si compared to nano-Si. Please compare the values between GBL and PC or EC based electrolytes.

6. As commented above, PC is known to show poor performance compared to EC. Also, FEC is critical for Si anodes. Thus, authors' claim on improved cycling performance using GBL + DEC + FEC electrolyte is not convincing unless it shows improved performance compared to EC + DEC + FEC. This is especially the case because 87.5 % of capacity retention after 100 cycles is considered fast capacity fade for most practical applications.

7. Line 285: What do authors mean by 'robust'? Simply high elastic modulus? LiF is ceramic type materials and while it has high elastic modulus, it is brittle. Please elaborate why authors claim LiF is robust, or mechanically beneficial.

8. Is the AFM indentation method conducted on only one location or multiple locations? The electrodes have high surface roughness while the indentation depth is extremely small. I expect the contact area would largely vary depending on the indentation location. Thus, a lot of AFM-based indentation studies conduct a number of indentation on a electrodes and report the scatter to make conclusions.

9. I understand AFM manufacturers such as Bruker claim that an operator can obtain various mechanical properties. However, their demonstraion is usually on well defined geometry and selected properties. In this case, the geometry and the material are extremely complex; the force-displacement curve is not so clean. This is why AFM indentation is criticized and often time not convinced. Some readers may consider the authors are deriving too much conclusions based on the noisy curves (Fig. 5a - elastic strain limit, thickness, modulus). I suggest following two additions in the

supporting materials to make the results more convincing. (1) Please provide mathematical descriptions how authors are deriving the mechanical properties from the curve. (2) Please conduct a simple AFM indentation on extruded acrylic (surface is not extremely smooth) and demonstrate the validity of the technique.

10. How does the measured mechanical properties of SEI compare with previous reports?

The following is the detailed response to all comments from the Reviewers.

Reviewer #1:

The manuscript describes a novel concept regarding fine-tuning of solid-electrolyte-interphases by dissolving presumably harmful components by GBL-solvent. The concept is certainly novel and highly interesting and makes a step towards purposely designed SEI-layers for improved long term performance of silicon based electrodes. Nevertheless, a publication in Nature Communications cannot be recommended, because too many questions remain unclear and the assumption that those specific components in the SEI are actually harmful is far fetched. Moreover, many uncertainties regarding the impact on the cycle life remain unclear and the main hypothesis is not well enough supported. As one example, I could not find reference spectra of the identified "un-desired" components of the SEI dissolved in GBL. Saying this, the work is of importance to the scientific community and a lot of interesting results are in the manuscript, which deserve publication in a more specialized journal focusing on electrochemistry and batteries.

Response to Comment: Thank you for the positive comments on the novelty and the significance of our work. We would like to respond to the questions and comments below.

Regarding “the assumption that those specific components in the SEI are actually harmful is far fetched.”

In the manuscript, we did not state that specific components in the SEI are “harmful”. Instead, we consider some SEI components (e.g., LEDC, LEC, $\text{Li}_x\text{PF}_y\text{O}_z$ and other oligomers) are “undesired” or “unfavored”. The term "undesired" here is limited to a problem-specific, relative concept. Specifically, the interphase instability of the micron Si anode is mainly due to the mechanical failure caused by the volume change.¹ Some SEI components (e.g., LiF) with good mechanical properties can survive repeated volume changes. However, other components (e.g., LEDC) with poor mechanical properties easily degrade and lead to disintegration of SEI, which

have therefore been defined as the *undesired* species. In our work, typical alkyl carbonates including LEDC and LEC were concerned as the undesired components, which have been extensively demonstrated with weak mechanical properties.²⁻⁴ Converting the fragile LEDC to the robust PEO type polymer in SEI has also been confirmed to promise an enhanced cycling performance of Si-based anode.² **Thus, we considered these components to be undesired or unfavored in terms of the mechanical property, but not “harmful”.** We have modified related discussions in the Revised Manuscript to make this point clear (see Page 4, Lines 66-70).

“Depending on the resilience of these components, which is correlated to the elastic modulus and strain limit, some components with high modulus (such as LiF) are desired to the mechanical strength and interface stability of the Si anode, whereas some with low resilience (either low modulus or low strain limit) are undesired and adverse to the mechanical properties of the SEI (see Supplementary Note 1 and Supplementary Table 1).”

Regarding “many uncertainties regarding the impact on the cycle life remain unclear”

The main problem of the cycling performance of silicon anodes is numerous secondary negative effects triggered by volume change, such as electrical isolations of active materials, unstable interphase, active Li loss.⁵ In this work, we aim to solve the interphase issue on the silicon anodes by designing the electrolyte and tailoring the SEI composition. The other factors such as anode materials are kept the same in the experimental and control samples and thus do not affect the comparison results and conclusions.

Regarding “the main hypothesis is not well enough supported”

The main hypothesis is the GBL can selectively dissolve the SEI species and form the SD-SEI with designed species. We have demonstrated this by (1) directly confirming the dissolved unfavored SEI components in GBL solvent and (2) characterizing the undissolved SEI components left on the soaked Si anodes.

Specifically, we have conducted a series of characterizations to probe the dissolution of the undesired SEI components. The LEDC dissolved from the cycled Si anode by GBL solvent can be detected via NMR analysis (Fig. R1). This result has

been also confirmed by the damaged SEI of the soaked Si anode in Fig. 1c and the enhanced ion conductivity of the GBL solvent in Supplementary Fig. 1. The undissolved LiF was also been detected on the soaked Si anode by the XPS in Supplementary Fig. 3. The SD-SEI with designed components showed improved mechanical properties and enhanced cycling performance, demonstrated by the following characterizations in Fig. 4 and electrochemical measurement in Fig. 3. We have modified related discussions in the Revised Manuscript to make this point clear (see Page 5, Lines 96-98).

“The selective dissolution capability of GBL were demonstrated by (1) directly confirming the dissolved unfavored SEI components in GBL solvent and (2) characterizing the undissolved SEI components left on the soaked Si anodes.”

Fig. R1 | Components analysis. NMR spectra of the pure GBL solvent and the GBL-based electrolyte collected from the cycled Si||NCM811 cell.

Regarding “reference spectra of the identified “un-desired” components of the SEI dissolved in GBL”

The undesired SEI components, such as LEDC, was directly determined by NMR spectra in Fig. R1. The “undesirable” property of LEDC (and other alkyl carbonates, oligomers, and other inorganics) here is relative to the LiF and polycarbonates, as these species have poor resilience (U_{max}) comparable to LiF and polycarbonates. The resilience (U_{max}) refers to the deformation energy within the whole elastic zone (that

is, maximum elastic strain energy), correlated to elastic modulus and elastic strain limit, defined by^{6,7} :

$$U_{max} = \frac{8}{15} \left(\frac{4}{5} \pi \right)^5 (1 - \nu^2)^4 E \varepsilon_Y^5 r^3$$

where ν is the SEI Poisson's ratio, E is elastic modulus, ε_Y is elastic strain limit and the r radius of the rigid indenter. According to previous work⁸, the ROCO₂Li and ROLi were fragile with Young's modulus less than 1 GPa, which is much lower than that of LiF (Table R1).

Table R1 | Comparison of modulus and elastic stain limit of typical SEI components.

	LiF	polycarbonates	ROCO ₂ Li	ROLi
Modulus (E, GPa)	89.6	<1	<1	<1
Elastic strain limit (ε_Y)	low	high	low	low

Considering their poor elasticity, these small-molecule species are less likely to have resilience comparable to LiF and polycarbonates⁷, and were therefore considered as “undesired” SEI components. As a predominant SEI component in EC-based electrolytes, the LEDC suffered from mechanical degradation and the chemical decomposition, which explain the inferior stability of LEDC-rich SEI on Si-based anodes.⁹ We have added related discussions in the Revised Manuscript (see Page 4, Line 69-70) and Supplementary Note 1 to clarify the meaning of the “undesired” components with references.

“...is crucial for the interfacial stability of the Si-based anode (see Supplementary Note 1).”

Reviewer#2:

Comment 1 : • What are the noteworthy results?

- The high capacity and good capacity retention are impressive for a micron-size Si-based anode. Also, the proposed method is evidently effective in modifying/regulating the SEI.

Response to Comment 1: Thank you for the positive comments.

Comment 2 : • Will the work be of significance to the field and related fields? How does it compare to the established literature? If the work is not original, please provide relevant references.

- One of the biggest hurdles in the Si community is achieving a high capacity with good capacity retention and the work presents a novel approach to address this by in-situ regulating the SEI composition.

Response to Comment 2: Thank you for the positive comments.

Comment 3 : • Does the work support the conclusions and claims, or is additional evidence needed?

- The authors have performed a comprehensive characterization of the SEI, combining computational and experimental techniques. The main claim is that the improved performance is primarily due to an SEI with good mechanical properties, achieved by dissolving the unwanted components. However, it is highly possible that the higher capacity is in part, if not mainly, due to a thinner SD-SEI compared to F-SEI or c-SEI. For instance, Fig. 1c shows more Si for the sample soaked in GBL, which is a good indication of a thinner SEI for this system. EIS, although shown only after 200 cycles, may also be an indication of a thinner SD-SEI. It would be good if the authors can explore this aspect in their discussion.

Response to Comment 3: Thank you for the positive general comments and the inspirations. We note that the samples in Fig. 1c were not SD-SEI but c-SEI after being soaked by various solvents including GBL, PC and EC/DEC. The SD-SEI was

analyzed by the ToF-SIMS analysis in Fig. 2f-h and force-displacement curves in Fig. 4d-f, where SD-SEI was proved to be thinner than the c-SEI but thicker than the F-SEI. Selective dissolution processes do not necessarily result in thinner SEI. The relationship between the specific capacity of Si anode and thickness of SEI may change with the cycling.

Inspired by the Reviewer's comment, **we believe it is reasonable to attribute the enhanced specific capacity to the improved charge transfer properties of SD-SEI.** The EIS result of cycled Si anodes directly revealed a lowered interphase charge-transfer resistance of SD-SEI compared to that of F-SEI and c-SEI. Therefore, at the same cathodic cut-off voltage, a reduced SEI resistance results in a lower practical electrode potential and consequently a higher state of charge (SOC). Therefore, the SD-SEI with reduced impedance can enable an enhanced specific capacity performance of Si anodes, as shown in the rate performance in Fig. 3c.

We have provided the related discussion in the Revised Manuscript to make this point clear (see Page 13, Lines 234-237; Page 14, Line 256).

“The typical charge/discharge profiles of the micron-sized Si anodes in the GBL-base electrolyte in Fig. 3a exhibit an initial capacity of 3307.2 mA h g⁻¹ in GBL-based electrolyte, higher than that of Si anodes in PC-based and EC-based electrolytes (Supplementary Fig. 18), likely due to the improved charge transfer properties of SD-SEI.”

“...which confirmed the improved charge transfer properties of SD-SEI.”

Comment 4: - In relation to the comments above, SD-SEI presents the highest initial capacity. Is this attributed to the mechanical property or to other characteristics of the SEI? A mechanically robust SEI does not necessarily translate to the accessible capacity of Si especially in the beginning of cycling.

Response to Comment 4: We agree with the reviewer that a robust mechanical property of SEI does not necessarily leads to improved capacity of Si anodes (especially initial capacity). **We attribute the increased specific capacity of Si anode with SD-SEI to the reduced SEI resistance instead of its mechanical**

properties, which has also been indicated by the rate performance in Fig. 3c. A similar situation is also found in the previous literature¹⁰, where a small SEI resistance has also been regarded as the main reason for the high specific capacity during the initial cycling test and enhanced rate performance. The enhanced mechanical properties of SD-SEI are beneficial to the interphase stability and the consequently enhanced capacity retention of micron-sized Si anode during the cycling compared with that of F-SEI and c-SEI.

Comment 5: - The mechanical aspect of the SEI may indeed be responsible for the cycling stability for SD-SEI. I believe that it is necessary to show characterization at various points upon cycling, e.g. cycle 1, cycle, 10, cycle 50, etc., to show that the SEI properties, particularly the mechanical integrity is indeed maintained upon cycling.

Response to Comment 5: Thank you for the valuable suggestions. According to Reviewer's suggestions, we have conducted XPS and AFM characterizations for the Si anode after different cycles to investigate the SEI involution and its mechanical integrity during cycling.

Fig. R2 displays the XPS spectra of the SEI on the Si anodes after 1st, 10th, and 50th cycles, based on which the Fig. R3 reveal the component evolution of SD-SEI upon cycling. While the intensity of LiF is high at 1st cycle and slightly decreased after cycling, the polycarbonates increase and level off after 10th cycle. The mechanical properties also evidenced above characteristics in components. According to Fig. R4-5, the Derjaguin–Muller–Toporov (DMT) modulus of SD-SEI at 20th cycle is lower than that at 1st and 10th cycle, which remained stable after 50th cycle. These results confirmed that the SD-SEI was gradually stabilized after 10 cycles. Based on the standard deviation of DMT modulus in Fig. R5, the mechanical integrity of the SD-SEI was well preserved and became more stable during the cycling.

We have added these post-cycling characterizations and analyses to the Revised Manuscript and SI (see Page 16, Lines 300-304)

“As confirmed by the modulus evolution collected in Supplementary Figs. 28-29, the

modulus of SD-SEI is stable during cycling test, indicating its enhanced mechanical properties and stable structure.”

Fig. R2 | Components analysis. XPS spectra of SD-SEI in various cycling number.

Fig. R3 | Components analysis. Relative content of selected components in the various SEI in different cycle number.

Fig. R4 | Mechanical properties after cycling. Derjaguin–Muller–Toporov (DMT) modulus mappings of SD-SEI of the Si anodes after (a) 1st, (b) 10th, (c) 20th and (d) 50th cycle.

Fig. R5 | Mechanical properties after cycling. Comparison of modulus and corresponding standard deviation of SD-SEI after different cycling numbers.

Comment 6: - It has been mentioned that the SD-SEI is compact. A surface image should be added to support this claim.

Response to Comment 6: The surface image of SD-SEI was obtained based on the

AFM. According to the 3D AFM images in Fig. R6, the SD-SEI was relatively flat and uniformly coated on the Si anode surface. The morphology of micron-sized Si anode beneath the SD-SEI was maintained, indicating a compact SD-SEI that well defend Si anode in contrast to the F-SEI and the c-SEI (Fig. R6b-c). Moreover, the proportional distribution of the SEI modulus in Fig. 6 was shown in Fig. R7. In sharp contrast to the c-SEI, the SD-SEI exhibited a single sharp peak (corresponding standard deviation is 0.279 log(Pa), 0.273 log(Pa) and 1.04 log(Pa)) for the SD-SEI, the F-SEI and the c-SEI, respectively), further indicating an a compact and homogeneous SD-SEI formed on the Si anodes.

We have provided the related discussion in the Revised Manuscript to make this point clear (see Page 16, Lines 296-297).

“According to the atomic force microscopy (AFM), the SD-SEI is rather flat and compact in contrast to F-SEI and c-SEI (Supplementary Figs. 26-27).”

Fig. R6 | Morphology analysis. 3D morphology images of cyclized micron-sized Si anode with (a) SD-SEI, (b) F-SEI and (c) c-SEI based on AFM.

Fig. R7 | Modulus Analysis. Modulus distribution of various SEIs.

Comment 7: - Do these prohibit publication or require revision?

- The discussion emphasizes early on that LiF is a beneficial SEI component, giving the impression that the more LiF the better the SEI. However, F-SEI has the most amount of LiF but has a much inferior performance, due to the brittle and fragile nature of its SEI. On the other hand, SD-SEI has the most desirable SEI with moderate amount of LiF and polycarbonates, resulting in a tough and mechanically robust SEI. It should be highlighted that it is potentially not the absolute amount of LiF in the SEI that is important, rather its amount relative to other species such as polymeric species in the case of SD-SEI. For instance, what if the same amount of LiF is left (relative to a control) after dissolution of unwanted SEI components; would the modified SEI still perform better?

Response to Comment 7: We thank the reviewer for the comments and instructive suggestions. We wish to clarify that LiF is a beneficial SEI component, but it is not “the more, the better”, because LiF is a brittle inorganic component, which has been widely discussed in literature. As we demonstrated in the manuscript, both high-modulus LiF and elastic polycarbonates are both important mechanically stable components and together construct a robust SEI. As shown in Fig. 3b, the SD-SEI with both LiF and polycarbonates enables better interfacial stability compared to the F-SEI (with 75.5% LiF based on the result from XPS), which highlights the synergistic effect of LiF and polycarbonates over the high-LiF-content F-SEI. Therefore, we conclude that the interfacial stability is closely related to the coexisting of LiF and polycarbonates. Just as the Reviewer commented, it is not the absolute amount but the relative amount of LiF in the SEI (containing polymeric species) that is critical to the stability of SEI.

We further demonstrate this point by changing the content of FEC in the GBL-based electrolyte to tune the relative content of LiF in the SEI. The GBL-based electrolytes with different FEC contents were denoted as GBL-xFEC electrolyte, where the “x” represents the volume ratio of FEC. Fig. R8 shows the SD-SEI formed in GBL-based electrolytes with different FEC contents. As expected, the lower

content of FEC leads to decreased amount of LiF of SD-SEI in the GBL-5FEC electrolyte and GBL-0FEC electrolyte. The cycling performance of micron-sized Si anode in the GBL-10FEC electrolyte (i.e., GBL-based electrolyte, in which the FEC content was 10% by volume) was found to be better than that in the GBL-5FEC electrolyte and GBL-0FEC electrolyte (Fig. R9 and Table R2). Therefore, **high relative content of LiF in SEI potentially benefits the cycle performance of Si anodes, and the synergistic effect of LiF and polycarbonates improve the cycle performance more significantly**. We have provided the related discussion in the Revised Manuscript to make this point clear (see Page 18, Lines 332-334).

“The synergistic effect between LiF and polycarbonates was also reflected by the decreased cycling stability of the micron-sized Si anode with SEI of lower LiF and polycarbonates content (Supplementary Figs. 35-36 and Supplementary Table 3).”

More related discussions were also added in Supplementary Figs. 35-36 in the Revised Supplementary Information.

Fig. R8 | Components analysis. XPS spectra of SD-SEI in GBL-based electrolytes with different FEC contents.

Fig. R9 | Electrochemistry. Cycling performance of the micron-sized Si anode in the GBL-based electrolytes with different content of FEC (forming different content of LiF).

Table R2 | SEI with various component configurations and corresponding cycling performance of Si anodes in GBL-mFEC electrolytes.

SEI (Electrolyte)	LiF Content (at %)	Polycarbonates Content (at %)	Capacity Retention After 50 Cycles (%)
SD-SEI (GBL-0FEC)	14.92	1.89	75.2
SD-SEI (GBL-5FEC)	52.59	5.87	96.6
SD-SEI (GBL-10FEC)	65.8	6.57	102.2
F-SEI (PC-10FEC)	75.5	~0	97.2

Comment 8: • Is the methodology sound? Does the work meet the expected standards in your field?

- The work is supported by several methods to support the claims made which therefore meet the expected standards.

Response to Comment 8: Thanks for the positive general comment.

Comment 9: • Is there enough detail provided in the methods for the work to be reproduced?

- Yes, there is enough details provided in the methods section. However, there is no mention regarding the reproducibility, particularly the electrochemical performance data.

Response to Comment 9: We thank the reviewer for the valuable suggestions. We now provide the data (Fig. R10) regarding the reproducibility of the electrochemical performance in Fig. 3. Fig. R10 are plotted based on the results from three sets of replicate experiments and the corresponding standard deviations. We have provided the related discussion in the Revised Manuscript to make this point clear (see Page 15, Lines 286-290).

“These above electrochemical performances, which exhibited good reproducibility (Supplementary Fig. 23), not only validate the feasibility of GBL-based electrolyte as a practical candidate to couple with Si-based anode materials but also support our idea of rational design of electrolyte and SEI for Si anodes.”

Fig. R10 | Cycling performance of micron-sized Si electrodes in Li||Si coin cells and Si||NCM811 full-cells. a, Initial charge/discharge profile of the micron-sized Si anode in the GBL-based electrolyte. **b,** Rate performance comparison. **c,** Cycling

performance and CEs of the micron-sized Si anode in various electrolytes. **d-e**, Initial charge/discharge profiles (**d**) and cycling performance I of the Si||NCM811 full-cells using various electrolytes.

For Reviewer #3:

General Comment: Mechanical issues surrounding SEI formed on Si are inevitable and critical for longevity of Si-based electrodes. Their concept of modifying physical properties of SEI by selectively tuning 'EI's components via electrolyte additive (GBL) are highly intriguing and expected to be practical. The authors provided results from a variety of characterizations to demonstrate the advantage of their approach. However, I have several questions and concerns around their characterization approaches and deriving conclusions, especially around 'EI's mechanical aspects. Since the core advantage of their approach is improved mechanical stability of SEI, I hope the authors properly address my concerns before acceptance for publication in Nature Comm. Below are my questions.

Response to Comment: Thanks for the positive general comments.

Comment 1: Authors are using three types of electrolytes. i) GBL + DEC + FEC, ii) PC + DEC + FEC, iii) EC + DEC. FEC is widely known to improve the performance of Si-based anodes. Why FEC is excluded from the EC-based electrolyte? Since PC is known to form poor SEI, to truly show the advantage of GBL over conventional electrolyte formulations, authors should compare electrochemical performance of Si electrodes between GBL + DEC + FEC vs. EC + DEC + FEC.

Response to Comment 1:

(1) Thank you for the valuable suggestion (1) “Why FEC is excluded from the EC-based electrolyte?”

The basic EC-based electrolyte without FEC is widely used in commercial LIBs and forms classic SEI. The classic SEI features fragile and porous components with poor mechanical properties, which is in sharp contrast to the SD-SEI with designed components and outstanding mechanical properties. It is hence used in this work as the control sample.

(2) “PC is known to form poor SEI”

The impression that “PC is known to form poor SEI” is mainly established in the studies of graphite anode, rather than Si-based materials. According to previous studies, PC-based electrolytes with additives¹¹ or high concentrations electrolytes¹² can realize stable cycling of Si anodes. In this work, the PC-based electrolyte generates optimized SEI, which results in better interfacial stability.

(3) “authors should compare electrochemical performance of Si electrodes between GBL + DEC + FEC vs. EC + DEC + FEC”

We thank the reviewer for this constructive suggestion. In response to reviewer’s concerns, we also investigated the electrochemical performance of the “EC+DEC+FEC” electrolyte (1 M LiPF₆ in EC/DEC/FEC electrolyte (45:45:10 by volume)) in Fig. R11. According to the charge/discharge profiles, micron-sized Si anodes in EC+DEC+FEC electrolyte exhibited an initial capacity of 2890 mA h g⁻¹. However, the GBL-based electrolyte still enables superior rate and cycling performance of micron-sized Si anode to that in EC+DEC+FEC electrolyte. These results further demonstrated the excellent interphase stability of Si anode realized by the SD-SEI and the selective dissolution strategy over conventional electrolyte formulations.

Fig. R11 | Electrochemical performance of micron-sized Si electrodes in EC+DEC+FEC electrolyte. a, Typical charge/discharge profiles of the micron-sized Si anode in the EC+DEC+FEC electrolyte. **b,** Rate performance comparison. **c,**

Cycling performance and CEs of the micron-sized Si anode in various electrolytes.

We have provided the related discussion in the Revised Manuscript to make this point clear (see Page 9, Lines 184-186; Page 10, Lines 187-188; Page 23, Lines 426-427).

“In contrast, the F-SEI contains only a large amount of LiF, and the c-SEI exhibits accumulating organic species, which could not be removed even if 10% FEC (by volume) was incorporated into the EC-based electrolyte (denoted as the EC-FEC electrolyte, see XPS analysis in Supplementary Fig. 11). These results further highlight the selectively dissolution strategy in screening the SEI components.”

“The EC-FEC electrolyte was prepared by dissolving 1M LiPF₆ in mixture of EC/DEC/FEC (45:45:10 by volume).”

Comment 2: Line 117: How do the authors define 'poor-mechanical-property'? Elastic modulus? Fracture toughness? Elastic strain limit? Please provide further ground why authors consider the suggested compounds (LEDC etc.) are mechanical poor components along with either mechanical measurements or references to back it up. Also, compounds such as LEDC may be beneficial for the passivity of Si surface.

Response to Comment 2: We thank the reviewer for the constructive comments and questions. **The “mechanical-property” here refers to the contribution of a SEI component on SEI structural stability confronting the evolving interface of Si anode during cycling, which can be specified to the resilience.** The resilience (U_{max}) refers to the deformation energy within the whole elastic zone (that is, maximum elastic strain energy), correlated to elastic modulus (E) and elastic strain limit (ϵ_Y), defined by^{6,7} :

$$U_{max} = \frac{8}{15} \left(\frac{4}{5} \pi \right)^5 (1 - \nu^2)^4 E \epsilon_Y^5 r^3 \quad (1)$$

where ν is the SEI Poisson's ratio, and the r is the radius of the rigid indenter. High modulus LiF is a key SEI component facilitating stabilized interphase on the Si-based anodes.^{10,13,14} Additionally, polymeric species including polycarbonates are also prevailing SEI components for their advantages in elastic strain limit.^{15,16} These

characteristics lead to a high resilience of SEI. In contrast, other components such as typical alkyl carbonates (like LEDC and LEC) and inorganic species (such as $\text{Li}_x\text{PF}_y\text{O}_z$) are considered to have “poor” mechanical property due to their lower resilience (lower modulus) compared to LiF and polycarbonates.²⁻⁴

Notably, while LEDC passivating anode is beneficial, the volume change of the Si anode would significantly damage the surface passivating, which was well demonstrated by the lower stability of LEDC on Si-based anodes than on graphite anode⁹. Upon exfoliating or disintegrating of LEDC from the interface, the electrolyte easily reacts with the absorbed Si anodes, which account for the loss of active lithium and Coulombic efficiency. Hence ideal SEI on Si-based anodes should features both mechanical stability and electrochemical stability.

We have provided the related discussion in the Revised Manuscript and Supplementary Note 1 in the Revised Supplementary Information to make this point clear (see Page 4, Lines 66-70).

“Depending on the resilience of these components, which is correlated to the elastic modulus and strain limit, some components with high modulus (such as LiF) are desired to the mechanical strength and interface stability of the Si anode, whereas some with low resilience (either low modulus or low strain limit) are undesired and adverse to the mechanical properties of the SEI (see Supplementary Note 1 and Supplementary Table 1).”

Comment 3: Lines 161-190: Daniel Abraham in ANL published several articles including 'What Makes Fluoroethylene Carbonate Different?'. In his articles, FEC promotes formation of both LiF and polymeric species. Authors discuss in this part that dissolution induced by GBL is the origin of higher LiF and polymeric species in SD-SEI compared to c-SEI. However, based on Daniel Abraham's papers, I wonder whether the observation is from the existence of FEC in GBL-based electrolyte while no existence of FEC in EC-based electrolyte.

Response to Comment 3: We thank the reviewer for the comments. FEC is indeed an important electrolyte additive for Si anodes. In the GBL-based electrolyte, FEC

functions as the main precursor of LiF and polymeric species and the GBL dissolves other undesired species in the SEI. Therefore, the use of FEC and selective dissolution induced by GBL are both important for higher content of LiF and polymeric species in SD-SEI compared to c-SEI.

We used the EC-based electrolyte without FEC additive as an important prototype electrolyte to obtain classic SEI as a control sample. Following the reviewer's suggestion, we have also tested the performance using EC+DEC+FEC electrolyte for comparison. As shown in Fig. R12, although the incorporation of FEC increased the LiF content, the other species (e.g., $\text{Li}_x\text{PO}_y\text{F}_z$, ROLi) were still present and affected the mechanical properties of the SEI. As we respond to comment 1 of Reviewer 3# (Fig. R11), the introduction of FEC cannot enable stable cycling of the micron-sized Si anode in the EC-based electrolyte comparable to that in the GBL-based electrolyte. **The introduction of FEC benefits the interfacial stability of the silicon cathode, but the selective dissolution effect in the presence of GBL results in a much greater performance enhancement due to the combined effect of LiF (high modulus) and polycarbonates (high elastic).**

Fig. R12 | SEI chemical compositions of cycle Si anodes. XPS characterizations of the SEI on the micron-sized Si anode in (a-c) the EC+DEC+FEC electrolyte and the (d-f) GBL-based electrolyte after 20 cycles.

We have provided the related discussion in the Revised Manuscript to make this

point clear (see Page 14, Lines 260-263).

“The enhanced electrochemical performance of SD-SEI is owing to the selective dissolution strategy by GBL. In contrast, the incorporation of FEC without GBL cannot realized stable cycling performance in the EC-FEC electrolyte (Supplementary Fig. 21).”

Comment 4: Please provide references stating LEDC or other species are 'undesirable' and elaborate why they are undesirable.

Response to Comment 4: The “undesirable” property of LEDC (and other alkyl carbonates, oligomers, and other inorganics) here is relative to the LiF and polycarbonates, as these species are not as high-modulus as LiF or as elastic as polycarbonates.^{9,17} Converting the fragile LEDC to the robust PEO type polymer in SEI has also been confirmed to promise an enhanced cycling performance of Si-based anode.²

Table R1 | Comparison of modulus and elastic stain limit of typical SEI components.

	LiF	polycarbonates	ROCO ₂ Li	ROLi
Modulus (E, GPa)	89.6	<1	<1	<1
Elastic strain limit (ϵ_Y)	low	high	low	low

According to previous report⁸, the ROCO₂Li and ROLi were fragile with Young's modulus less than 1 GPa, which is much lower than that of LiF (Table R1). Considering their poor elasticity, these small-molecule species are less likely to exhibit resilience comparable to LiF and polycarbonates^{7,18}, and were therefore considered as “undesired” SEI components. As a predominant SEI component in EC-based electrolytes, the LEDC suffered mechanical degradation and the following decomposition, which explain the inferior stability of LEDC-rich SEI on Si-based anode.⁹ We have modified related discussions in the Revised Manuscript and supplementary note 1 in the Revised Supplementary Information to make this point clear (see Page 4, Lines 66-70).

“Depending on the resilience of these components, which is correlated to the elastic

modulus and strain limit, some components with high modulus (such as LiF) are desired to the mechanical strength and interface stability of the Si anode, whereas some with low resilience (either low modulus or low strain limit) are undesired and adverse to the mechanical properties of the SEI (see Supplementary Note 1 and Supplementary Table 1)."

Comment 5: i) Please explain why GBL yields higher specific capacity. The described advantage of GBL doesn't seem to directly coupled to capacity utilization. ii) Higher coulombic efficiency is from lower surface area of micron-sized Si compared to nano-Si. Please compare the values between GBL and PC or EC based electrolytes.

Response to Comment 5:

(i) The lowered resistance of SD-SEI account for higher specific capacity in the GBL-based electrolyte compared with other electrolytes. Specifically, the measured specific capacity is determined thermodynamically and kinetically. Within the same discharge cut-off voltage, a reduced SEI resistance (Supplementary Figure 19) account for small overpotential, which result in a lower practical electrode potential and consequently higher state of charge (SOC). Therefore, the SD-SEI with reduced impedance can enable an enhanced specific capacity of Si anodes, which was also demonstrated in Figure 3c.

We have provided the related discussion in the Revised Manuscript to make this point clear (see Page 13, Lines 236-237).

"...higher than that of Si anodes in PC-based and EC-based electrolytes (Supplementary Fig. 18), likely due to the improved charge transfer properties of SD-SEI"

(ii) We agree with the Reviewer that the initial Coulombic efficiency of micron-sized Si anodes is higher than its nano-sized counterparts (~90% vs. ~80%)¹⁹, which is the advantage and the motivation to use micron Si. In our work, the initial Coulombic efficiencies of micron-sized Si anodes are 88.6%, 90.3% and 88.3% in the GBL-based electrolyte, PC-based electrolytes and EC-based electrolytes, respectively. The relatively lower initial Coulombic efficiency of Si-anodes in the GBL based

electrolytes could be attributed to the selective dissolution process at the surface of anode.

We have provided the related discussion in the Revised Manuscript to make this point clear (see Page 12, Lines 237-241).

“The initial Coulombic efficiency of micron-sized Si anode in the GBL-based electrolyte was much higher than that of nano-sized one but lower than that in the PC-based electrolyte (90.3% and 88.3% for the PC and EC-based electrolyte, respectively), which can be attributed to the charge consumption during the selective dissolution process.”

Comment 6: As commented above, PC is known to show poor performance compared to EC. Also, FEC is critical for Si anodes. Thus, authors' claim on improved cycling performance using GBL + DEC + FEC electrolyte is not convincing unless it shows improved performance compared to EC + DEC + FEC. This is especially the case because 87.5 % of capacity retention after 100 cycles is considered fast capacity fade for most practical applications.

Response to Comment 6: In this manuscript, we proposed a strategy of regulating the mechanical properties of SEI on micron-sized Si anode. Hence, we focused on the SEI regulating process, the compositional and mechanical characteristics of resulting SD-SEI and the enhanced electrochemical performance. We studied the electrochemical performance of micron-sized Si anode in EC+DEC+FEC electrolyte (Fig. R11), and provided discussion in response to Comment 1 of Reviewer #3 (see Page 16 of the Response Letter). **The micron-sized Si anode in the GBL-based electrolyte exhibited superior cycling performance to that in the EC+DEC+FEC electrolyte.**

Regarding the capacity retention, we respectfully note that the 87.5% capacity retention was realized by the micron-sized raw silicon anode after 100 cycles. The undecorated surface of this Si anode with 7 μm diameter (far beyond the 150 nm critical size of Si) is extremely vulnerable,²⁰ hence well highlighting the mechanical characteristics of SEI. In comparison, many previous studies cannot match this

performance (compared to for example, less than 60% capacity retention after 100 cycles²¹). We further used with a practical anode, micron-sized Si@C anode, with the SD-SEI strategy and realized stable cycling performance over 500 cycles (Fig. 3d). To further investigate the practical performance of the GBL-based electrodes and the derived SD-SEI, a pouch cell was also assembled incorporating NCM811 cathodes and graphite-SiO_x@C composite anode (500 mA h g⁻¹). The pouch full cell exhibits stable cycling performance with 80% capacity retention after 350 cycles (Fig. R13), which indicating the superior practical prospect and feasibility of this strategy.

We have provided the related discussion in the Revised Manuscript to make this point clear (see Page 18, Lines 336-339).

“Furthermore, a pouch cell assembled coupling NCM811 cathode and graphite-SiO_x@C composite anode (500 mA h g⁻¹, denoted as G-SiO_x) exhibits stable cycling performance with 80% capacity retention after 350 cycles (Supplementary Fig. 38).”

Fig. R13 | Cycling performance of G-SiO_x||NCM811 pouch cell. The inset shows the optical image of the pouch cell.

Comment 7: Line 285: What do authors mean by 'robust'? Simply high elastic modulus? LiF is ceramic type materials and while it has high elastic modulus, it is brittle. Please elaborate why authors claim LiF is robust, or mechanically beneficial.

Response to Comment 7: We thank the Reviewer for carefully reviewing our manuscript and delivering helpful comments. Herein we attempt to highlight the high elastic modulus of LiF in combined with elastic polycarbonates, and the “robust LiF”

has been replaced by “stiff LiF” in the Revised Manuscript. The LiF is characterized by its high elastic modulus (89.6 GPa)⁸, thus promising high resilience (U_{\max} , see response to Comment 3 of Reviewer #3) and outstanding tolerance towards the evolving interphase.²²⁻²⁴ However, LiF alone is inelastic with small elastic strain limit (ϵ_Y). In the SD-SEI, the stiff LiF and elastic polycarbonates together provide superior elasticity and maintain the electrode integrity of the interphase.

We have modified related discussions to make this point clear (see Page 16, Lines 297-298).

“Interweaving with both highly elastic polycarbonates and stiff LiF, SD-SEI exhibits a suitable average modulus of 1.5 GPa (Fig. 4a).”

Comment 8: Is the AFM indentation method conducted on only one location or multiple locations? The electrodes have high surface roughness while the indentation depth is extremely small. I expect the contact area would largely vary depending on the indentation location. Thus, a lot of AFM-based indentation studies conduct a number of indentation on a electrodes and report the scatter to make conclusions.

Response to Comment 8: In response to the reviewer’s concerns, we further complemented the force-displacement curves collected from various locations of SEI. According to the results in Fig. R14, the elastic strain limit of SD-SEI is averagely 15 nm, which is much higher than that of F-SEI and c-SEI (~8 nm and 7 nm, respectively). This result is consistent with the conclusion in the manuscript (13.4 nm, 7.8 nm and 7.5 nm for SD-SEI, F-SEI and c-SEI, respectively), which further confirmed the validity of this measurement.

We have modified related discussions in the Revised Manuscript to make this point clear (see Page 17, Lines 306-308).

“As for the maximum elastic deformation limit, SD-SEI shows an overwhelming superiority to that of F-SEI and c-SEI (15 nm, 8 nm and 7 nm according to the average result shown in Supplementary Fig. 33).”

Fig. R14 | Mechanical properties analysis. **a-c.** 2D morphology images of (a) SD-SEI, (b) F-SEI and (c) c-SEI based on AFM. **d-f,** force-displacement curves of (d) SD-SEI, (e) F-SEI and (f) c-SEI collected from the red points in the corresponding morphology images.

Comment 9: I understand AFM manufacturers such as Bruker claim that an operator can obtain various mechanical properties. However, their demonstration is usually on well defined geometry and selected properties. In this case, the geometry and the material are extremely complex; the force-displacement curve is not so clean. This is why AFM indentation is criticized and often time not convinced. Some readers may consider the authors are deriving too much conclusions based on the noisy curves (Fig. 5a - elastic strain limit, thickness, modulus). I suggest following two additions in the supporting materials to make the results more convincing. (1) Please provide mathematical descriptions how authors are deriving the mechanical properties from the curve. (2) Please conduct a simple AFM indentation on extruded acrylic (surface is not extremely smooth) and demonstrate the validity of the technique.

Response to Comment 9: We thank the reviewer for the helpful comments. In the experiments, the validity of the force-displacement curves was ensured by the following points: (1) The data were collected in the relatively flat area, so nearly no torque was generated during the measurement. (2) The electrode, together with the attached micron-sized Si particles was compact during the cell assembly and the ensuing volume expansion process. Therefore, the pressure applied during the measurement is less likely to cause structural evolution, which provided a relatively stable environment for the measurement. (3) The sharp contrast between Si and SEI components (>150 GPa vs. ~1 GPa) further enhanced the accuracy of the measurement.

First, based on the Reviewer's comment, the process of nanoindentation is schematic illustrated in Fig. R15. According to the Hertz contact model, the relationship between applied force (F) and the indentation (δ) in the elastic deformation region of SEI (Fig. R16) can be described as followed:

$$F = \frac{4}{3} \frac{E}{1 - \nu^2} \sqrt{r} \delta^{3/2} \quad (2)$$

where E is the Young's modulus, ν is the Poisson's ratio, r is radius of the indenter. The Equation 2 was used to fit the obtained curves, as shown in Fig. R16 (black curve). The position of point was precisely located as the first deviation point from the fitting curve of Hertz contact model. Similarly, this model was applied again after indenter impaling the SEI and directly contacting the Si anode. All the force-displacement curves were analyzed as above.

Second, we conducted the indentation experiment on an acrylic plate to further valid the result from the force-displacement curves. According to the result, the Young's modulus of the acrylic plate was 2.2 GPa (Fig. R17), which agrees with result in the previous report (2.4 GPa²⁵).

We have provided the related discussion in the Revised Manuscript and Supplementary Note 4 in the Revised Supplementary Information to make this point clear (see Page 16, Lines 303-304).

"The nanoindentation test was also conducted to uncover the SEI evolution in

response to the stress (see Fig. 4d-f, Supplementary Note 4 and Supplementary Figs. 30-32).”

Fig. R15 | Schematic diagram of the force-displacement curve test process based on AFM, including (a) no contact, (b) elastic deformation, (c) plastic deformation of SEI, (d) penetrating the SEI layer and contacting the Si anode surface.

Fig. R16 | Fitting process of force-displacement curve.

Fig. R17 | Mechanical properties. **a**, Morphology of acrylic plate. **b**, Force-displacement curve on an acrylic plate.

Comment 10: How does the measured mechanical properties of SEI compare with previous reports?

Response to Comment 10: We thank the reviewer for the helpful questions. Table 3 shows the comparison of thickness, elastic strain limit (ϵ_Y) and modulus (E) of SEIs. Very few reports managed to measure the ϵ_Y . The SD-SEI exhibited advantages in terms of mechanical properties over previous reports. Noteworthy, as the results in different papers may be obtained by different methods or instruments with different parameters or conditions, this comparison may be not exactly fair.

Table 3 | Comparison of mechanical properties of SEIs in various reports.

Anode	Thickness (nm)	ϵ_Y (nm)	E (GPa)	Citation
Si	~24.31	\	~69.2	26
Si	25	\	3.7	27
Si	24	\	~0.696	28
Si	62	\	0.296	29
Si	145	\	0.095	
Si	14.5	\	2.4	30
Sn (SIBs)	~8	~4	0.355	31
Na (SIBs)	~12.9	~10.4	~1	32
Si	32.8	13.4	1.5	This work

We have provided the related discussion in the Revised Manuscript to make this point clear (see Page 18, Lines 326-329).

“Thus, the SD-SEI due to the selective dissolution of the high-DN GBL solvent affords optimized mechanical parameters, such as elastic strain limit, deformation-inhibiting capability, and other physicochemical properties, compared with F-SEI, c-SEI, and SEIs in previous reports (Fig. 5a and Supplementary Table 2)”

References

1. Jin, M. Y. *et al.* Optimum Particle Size in Silicon Electrodes Dictated by Chemomechanical Deformation of the SEI. *Adv. Funct. Mater.* **31**, 2010640 (2021).
2. Yang, Z. *et al.* In-situ cross-linking strategy for stabilizing the LEDC of the solid-electrolyte interphase in lithium-ion batteries. *Nano Energy* **105**, 107993 (2023).
3. Shin, H., Park, J., Han, S., Sastry, A. M. & Lu, W. Component-/structure-dependent elasticity of solid electrolyte interphase layer in Li-ion batteries: Experimental and computational studies. *J. Power Sources* **277**, 169–179 (2015).
4. Heiskanen, S. K., Kim, J. & Lucht, B. L. Generation and Evolution of the Solid Electrolyte Interphase of Lithium-Ion Batteries. *Joule* **3**, 2322–2333 (2019).
5. Chae, S., Ko, M., Kim, K., Ahn, K. & Cho, J. Confronting Issues of the Practical Implementation of Si Anode in High-Energy Lithium-Ion Batteries. *Joule* **1**, 47–60 (2017).
6. Gao, Y. & Zhang, B. Probing the Mechanically Stable Solid Electrolyte Interphase and the Implications in Design Strategies. *Adv. Mater.* **35**, 2205421 (2023).
7. Gao, Y. *et al.* Unraveling the mechanical origin of stable solid electrolyte interphase. *Joule* **5**, 1860–1872 (2021).
8. Wan, G. *et al.* Suppression of Dendritic Lithium Growth by in Situ Formation of a Chemically Stable and Mechanically Strong Solid Electrolyte Interphase. *ACS Appl. Mater. Interfaces* **10**, 593–601 (2018).
9. Kim, J., Chae, O. B. & Lucht, B. L. Perspective—Structure and Stability of the

- Solid Electrolyte Interphase on Silicon Anodes of Lithium-ion Batteries. *J. Electrochem. Soc.* **168**, 030521 (2021).
10. Chen, J. Electrolyte design for LiF-rich solid–electrolyte interfaces to enable high-performance micro-sized alloy anodes for batteries. *Nat. Energy* **5**, 386–397 (2020).
 11. Li, L. L. *et al.* Methyl phenyl bis-methoxydiethoxysilane as bi-functional additive to propylene carbonate-based electrolyte for lithium ion batteries. *Electrochimica Acta* **56**, 4858–4864 (2011).
 12. Chang, Z. *et al.* The Electrochemical Performance of Silicon Nanoparticles in Concentrated Electrolyte. *ChemSusChem* **11**, 1787–1796 (2018).
 13. Tan, J., Matz, J., Dong, P., Shen, J. & Ye, M. A Growing Appreciation for the Role of LiF in the Solid Electrolyte Interphase. *Adv. Energy Mater.* **11**, 2100046 (2021).
 14. Park, S. *et al.* Replacing conventional battery electrolyte additives with dioxolone derivatives for high-energy-density lithium-ion batteries. *Nat. Commun.* **12**, 838 (2021).
 15. Yang, G. *et al.* Robust Solid/Electrolyte Interphase (SEI) Formation on Si Anodes Using Glyme-Based Electrolytes. *ACS Energy Lett.* **6**, 1684–1693 (2021).
 16. Gao, Y. *et al.* Polymer–inorganic solid–electrolyte interphase for stable lithium metal batteries under lean electrolyte conditions. *Nat. Mater.* **18**, 384–389 (2019).
 17. Parimalam, B. S., MacIntosh, A. D., Kadam, R. & Lucht, B. L. Decomposition Reactions of Anode Solid Electrolyte Interphase (SEI) Components with LiPF₆. *J. Phys. Chem. C* **121**, 22733–22738 (2017).

18. Tsai, W.-Y., Thundat, T. & Nanda, J. Toward a mechanically stable solid electrolyte interphase. *Matter* **4**, 2119–2122 (2021).
19. Zhu, G., Chao, D., Xu, W., Wu, M. & Zhang, H. Microscale Silicon-Based Anodes: Fundamental Understanding and Industrial Prospects for Practical High-Energy Lithium-Ion Batteries. *ACS Nano* **15**, 15567–15593 (2021).
20. Liu, X. H. *et al.* Size-Dependent Fracture of Silicon Nanoparticles During Lithiation. *ACS Nano* **6**, 1522–1531 (2012).
21. Gu, L. *et al.* Enabling robust structural and interfacial stability of micron-Si anode toward high-performance liquid and solid-state lithium-ion batteries. *Energy Storage Materials* **52**, 547–561 (2022).
22. Fan, X. *et al.* Fluorinated solid electrolyte interphase enables highly reversible solid-state Li metal battery. *Sci. Adv.* **4**, eaau9245 (2018).
23. Chen, C. *et al.* Impact of dual-layer solid-electrolyte interphase inhomogeneities on early-stage defect formation in Si electrodes. *Nat. Commun.* **11**, 3283 (2020).
24. Wu, H., Jia, H., Wang, C., Zhang, J. & Xu, W. Recent Progress in Understanding Solid Electrolyte Interphase on Lithium Metal Anodes. *Adv. Energy Mater.* **11**, 2003092 (2021).
25. Kindt-Larsen, T., B. Smith, D. & Steen, J. J. Innovations in acrylic bone cement and application equipment. *J. Appl. Biomater.* **6**, 75–83 (1995).
26. Zhang, Q., Xiao, X., Zhou, W., Cheng, Y.-T. & Verbrugge, M. W. Toward High Cycle Efficiency of Silicon-Based Negative Electrodes by Designing the Solid Electrolyte Interphase. *Adv. Energy Mater.* **5**, 1401398 (2015).

27. Kamikawa, Y., Amezawa, K. & Terada, K. Elastic–Plastic Deformation of a Solid Electrolyte Interface Formed by Reduction of Fluoroethylene Carbonate: A Nanoindentation and Finite Element Analysis Study. *J. Phys. Chem. C* **124**, 22488–22495 (2020).
28. Huang, S. In-situ study of surface structure evolution of silicon anodes by electrochemical atomic force microscopy. *Appl. Surf. Sci.* **452**, 67–74 (2018).
29. Li, Y. *et al.* Suppressing Continuous Volume Expansion of Si Nanoparticles by an Artificial Solid Electrolyte Interphase for High-Performance Lithium-Ion Batteries. *Appl. Surf. Sci.* **9**, 8059–8068 (2021).
30. Cao, Z., Zheng, X., Qu, Q., Huang, Y. & Zheng, H. Electrolyte Design Enabling a High-Safety and High-Performance Si Anode with a Tailored Electrode–Electrolyte Interphase. *Adv. Mater.* **33**, 2103178 (2021).
31. Huang, J. *et al.* Nanostructures of solid electrolyte interphases and their consequences for microsized Sn anodes in sodium ion batteries. *Energy Environ. Sci.* **12**, 1550–1557 (2019).
32. Gong, C. *et al.* The role of an elastic interphase in suppressing gas evolution and promoting uniform electroplating in sodium metal anodes. *Energy Environ. Sci.* **16**, 535–545 (2023).

Reviewer #2 (Remarks to the Author):

The manuscript highlights an approach for stabilizing the SEI which is very critical for Si-based electrodes. I appreciate the authors' efforts in making all the additional experiments to satisfactorily answer the questions from the reviewers. Here are few more minor comments/questions to consider before recommending for publication in Nature Communications:

1. It would be useful to add frequencies in selected regions in the EIS plots in Fig. 20 (Supporting Information).

2. What is responsible for the lower charge transfer resistance of SD-SEI? How do you distinguish between the charge transfer and SEI resistance?

3. Regarding the impact of PC on the SEI of Si electrodes, you may want to check this reference (<https://doi.org/10.1016/j.jpowsour.2022.231021>) which evaluated the role of electrolyte components on the passivation of silicon electrodes and found the beneficial effects of PC.

Reviewer #3 (Remarks to the Author):

I carefully went through the authors' response to my comments. I consider the additional discussions and experimental results were sufficiently provided to address all of my concerns.

Thank you for the good work!

The following is the detailed response to all comments from the reviewers.

Reviewer #2:

The manuscript highlights an approach for stabilizing the SEI which is very critical for Si-based electrodes. I appreciate the authors' efforts in making all the additional experiments to satisfactorily answer the questions from the reviewers. Here are few more minor comments/questions to consider before recommending for publication in Nature Communications:

Response to Comment: Thank you for the positive comments on our revision. We would like to respond to the questions and comments below.

1. It would be useful to add frequencies in selected regions in the EIS plots in Fig. 20 (Supporting Information).

Response to Comment 1: We thank the reviewer for the instructive suggestion. The frequency information has been added in the Revised Supplementary Figure 20, as shown below.

Figure R1 | EIS analysis. Nyquist plots of Li||Si half-cell with GBL-based electrolyte, PC-based electrolyte and EC-based electrolyte in pristine state (a) and after 200 cycles (b). The corresponding equivalent circuit applied in pristine state (c) and cycled state (d). R_s , R_{sei} and R_{ct} represent the solution resistance, SEI resistance and charge transfer resistance, respectively. CPE, CPE1 and CPE2 stands for the relative double-layer capacitance. W_s represents the Warburg impedance related to lithium-ions diffusing. The fitting results were plotted as the solid curves in (a) and (b).

2. What is responsible for the lower charge transfer resistance of SD-SEI? How do you distinguish between the charge transfer and SEI resistance?

Response to Comment 2: We thank the reviewer for the questions. The SD-SEI has lower resistance because it is composed of low-resistance species and has better chemical and mechanical stability, compared with F-SEI and c-SEI. Specifically, in the SD-SEI, high-resistance organic SEI species were selectively dissolved. Additionally, the robust SD-SEI can survive the volume change of the Si anode and thus prevent the exposure of fresh Si interface and excessive generation of SEI (which would increase interface resistance). The charge transfer resistance is accordingly lowered owing to the well-protected Si anode against electrolyte erosion.

The charge transfer resistance (R_{ct}) represents the impediment to current during charge transfer process in an electrochemical reaction, while the SEI resistance (R_{SEI}) originated from the SEI. According to previous report¹, the Nyquist plot of cycled anode can be deconvoluted into two semi-cycles of different radius and linear region. The semi-cycles located at higher frequency indicates the R_{SEI} , while the other one at lower frequency represents the R_{ct} . **Therefore, we distinguished the R_{ct} and R_{SEI} by their frequencies.** Furthermore, the specific value of R_{SEI} and R_{ct} can be determined by the fitted results obtained from the equivalent circuit in Figures R1d. Compared with that of R_{SEI} (1.6 Ω , 67.6 Ω and 193.5 Ω for SD-SEI, F-SEI and c-SEI), R_{ct} exhibits higher value (4.7 Ω , 147 Ω and 407 Ω for SD-SEI, F-SEI and c-SEI).

To make this point clear, we have modified the related discussion in the Revised Manuscript (see Page 13, Line 253; Page 14, Lines 254-256) and Revised

Supplementary Information Fig. 20 (see Page S26, Lines 233-237).

“The EIS spectra before and after 200 cycles also confirm a smaller voltage polarization of Li||Si cell with GBL electrolyte compared with PC-based electrolyte and EC-based electrolyte (Supplementary Fig. 20), which confirmed the improved kinetics properties of SD-SEI.”

“The semi-cycles located at higher frequency indicates the R_{SEI} , while the other one at lower frequency represents the R_{ct} . In terms of R_{SEI} , the SD-SEI shows lower value compared with that of F-SEI and c-SEI (1.6 Ω vs. 67.6 Ω vs. 193.5 Ω for SD-SEI, F-SEI and c-SEI), indicating the favored kinetics properties and excellent stability of SD-SEI.”

3. Regarding the impact of PC on the SEI of Si electrodes, you may want to check this reference (<https://doi.org/10.1016/j.jpowsour.2022.231021>) which evaluated the role of electrolyte components on the passivation of silicon electrodes and found the beneficial effects of PC.

Response to Comment 3: We thank the reviewer for the valuable suggestion. We have carefully read this reference. This report highlighted the positive role of PC in building more stable SEI on the Si anode compared to that of EC, which support our experimental design. We have cited this work in the Revised Manuscript (see Page 4, Lines 66, Citation 25).

“Nevertheless, the native SEI are complex, usually containing inorganic components (e.g., LiF, LiOH, and Li₂CO₃) and organic components (e.g., various lithium alkyl carbonates and polycarbonates)^{24,25}.”

Reviewer #3:

I carefully went through the authors' response to my comments. I consider the additional discussions and experimental results were sufficiently provided to address all of my concerns. Thank you for the good work!

Response to Comment: Thank you for the positive comments on our revision.

References

1. Qu, D., Ji, W. & Qu, H. Probing process kinetics in batteries with electrochemical impedance spectroscopy. *Commun. Mater.* **3**, 61 (2022).